

# Super-cooled liquid fogs over the central Greenland ice sheet

Christopher J. Cox[1,2], David C. Noone[3], Max Berkelhammer[4], Matthew D. Shupe[1,2], William D. Neff[1,2], Nathaniel B. Miller[1], Von P. Walden[5], Konrad Steffen[6]

[1] Cooperative Institute for Research in Environmental Sciences, Boulder, Colorado, 80309, USA
[2] NOAA Earth System Research Laboratory, Boulder, Colorado, 80305, USA
[3] College of Earth, Ocean, and Atmospheric Sciences, Oregon State University, Corvallis, Oregon, 97331, USA
[4] Department of Earth and Environmental Sciences, University of Illinois at Chicago, Chicago, Illinois, 60607, USA
[5] Department of Civil and Environmental Engineering, Washington State University, Pullman, Washington, 99164, USA
[6] Swiss Federal Research Institute WSL, Birmensdorf, CH-8903, Switzerland

*Correspondence to*: Christopher J. Cox (christopher.j.cox@noaa.gov)

**Abstract.** Radiation fogs at Summit, Greenland (72.58°N, 38.48°W, 3210 masl) are frequently reported by observers. The fogs are often accompanied by fogbows, indicating the particles are composed of liquid and because of the low temperatures at Summit, this liquid is super-cooled. Here we analyse the formation of these fogs as well as their physical and radiative properties. In situ observations of particle size and droplet number concentration were made using scattering spectrometers near 2 m and 10 m height from 2012 to 2014. These data are complemented by co-located observations of meteorology, turbulent and radiative fluxes, and remote sensing. We find that liquid fogs occur in all seasons with the highest frequency in September and a minimum in April. Due to the characteristics of the boundary-layer meteorology, the fogs are elevated, forming between 2 m and 10 m and the particles then fall toward the surface. The diameter of mature particles is typically 20-25 μm in summer. Number concentrations are higher at warmer temperatures and, thus, higher in summer compared to winter. The fogs form at temperatures as warm as warm as -5 °C, while the coldest form at temperatures approaching -40 °C. Facilitated by the elevated condensation, in winter 2/3 of fogs occurred within a relatively warm layer above the surface when the near-surface air is below -40 °C, as cold as -57 °C, which is well below that which can support liquid water. This implies that fog particles settling through this layer of cold air freeze in the air column before contacting the surface, thereby accumulating at the surface as ice without riming. Liquid fogs under otherwise clear skies impart annually 1.5 W m² of cloud radiative forcing (CRF). While this is a relatively small contribution to the surface radiation climatology, individual events are influential. The mean CRF during liquid fog events is 26 W m², but can sometimes be much higher. An extreme case study was observed to radiatively force 5 °C of surface warming during the coldest part of the day, effectively damping the diurnal cycle. At lower elevations of the ice sheet where melting is more common, such damping could signal a role for fogs in preconditioning the surface for melting later in the day.



# 1 Introduction

Fogs are reported by observers at Summit, Greenland (72.58°N, 38.48°W, 3210 masl) approximately 18% of the time in autumn and 8-10% of the time in other months (Starkweather, 2004). In sunlight, these fogs are at times accompanied by characteristic fogbows, the presence of which suggests the fog is optically thin and therefore transmits solar radiation, and that the particles are spherical, indicating that the fog is composed of liquid. Since Summit is situated in the ice sheet accumulation zone, a region that rarely experiences temperatures above freezing (Nghiem et al., 2012), the liquid in fogs observed there is super-cooled. Persistent and strong surface-based temperature inversions occur throughout the year (Miller et al. 2013). The cooling associated with the development of the inversions drives saturation in the atmospheric boundary-layer and produces the fog condensate (e.g., Bergin et al., 1995; Hoch et al., 2007; Berkelhammer et al., 2016). Cooling rates during fog events have been observed to be up to -35 K $d^{-1}$ within the lowest 50 m (Hoch et al. 2007). Due to seasonal differences in the vapor mixing ratio gradient, the moisture more likely originates from the free atmosphere in summer, while in winter when the boundary-layer is decoupled from the free troposphere, moisture is more likely to be recycled within a few meters above the surface through a process involving sublimation, condensation and settling (Berkelhammer et al., 2016). Meteorology is insufficient to explain fog presence (Tjernström, 2005) and thus, other parameters such as aerosols (Bergin et al. 1995), turbulent fluxes (Gultepe et al., 2007; Hoch et al., 2007; Berkelhammer et al., 2016) and dynamics (Nakanishi 2000) are necessary to understand the processes that govern fog development and the hydrological and energetic interactions that fogs have with the surface.

Fog microphysics influence their radiative properties. For example, liquid more efficiently absorbs and emits longwave radiation than ice (e.g., Garrett and Zhao, 2006; Shupe and Intrieri, 2004), likely increasing the forcing from liquid fog per unit mass compared to ice fogs or clear-sky ice precipitation ("diamond dust", Intrieri and Shupe, 2004). Despite being optically thin, fogs at Summit have been reported to increase the downwelling longwave flux by up to 20 W $m^{-2}$ in summer and 75 W $m^{-2}$ in winter (Starkweather, 2004). They have also been linked to reduced aerosol loading through surface riming (Borys et al., 1992; Bergin et al. 1995) and to limiting ice sheet accumulation loss via sublimation in the decoupled wintertime state (Berkelhammer et al., 2016). In summer, radiative processes associated with optically thin tropospheric clouds can influence surface melt (Bennartz et al., 2013), a mechanism that could plausibly pertain to fogs as well.

Fogs observed at Summit occur within a shallow layer above the surface, just a few tens of meters thick (Berkelhammer et al., 2016). Consequently, fogs are likely under-represented by cloud climatologies because occurrence estimates from surface observations typically rely on lidar and radar measurements (e.g., Shupe et al., 2011), which are insensitive in the lowest hundred meters of the atmosphere. Satellite-based studies may also miss them because they are difficult to distinguish from the surface, which has a similar temperature (Crane and Anderson, 1984). Since the relevant processes occur at scales smaller than the vertical spacing of levels in climate models, the models are unlikely to resolve them. Despite these potential omissions, the coupled hydrological and energetic processes associated with fogs could have important implications for monitoring and projecting ice sheet surface mass balance and properly calibrating paleoclimate records





derived from ice cores. Additionally, the cold temperatures and persistent stable-stratification of central Greenland make Summit a useful location for fog process studies, in particular as examples in meteorological extremes for informing model development of fogs, the forecasts for which are important for aviation and transportation safety. Furthermore, shallow fogs afford opportunities to study the evolution of clouds in situ for extended periods, which contributes to broader studies of cloud

physics and cloud-aerosol interactions. Therefore, a focused effort on Greenland fog processes is warranted, building on previous studies (Borys et al., 1992; Bergin et al., 1995; Starkweather, 2004; Hoch et al., 2007; Berkelhammer et al., 2016).

From June 2012 through June 2014, measurements were made on and near a 46 m tower by the Closing the Isotope Balance at Summit (CIBS) experiment, in collaboration with personnel from ETH Zürich, who maintained the tower and nearby broadband radiometric measurements. The CIBS suite included light-scattering spectrometers (DMT Fog Monitors,

"FM100") mounted on the tower close to 2 m and 10 m alongside measurements suitable for deriving turbulent heat fluxes. The FM100 probes made in situ observations of near-surface hydrometeors between 1 and 50 μm. These data were collected adjacent to the Integrated Characterization of Energy, Clouds, Atmospheric state and Precipitation at Summit (ICECAPS) atmospheric observatory (Shupe et al., 2013), which operates ground-based remote sensor instruments for cloud and tropospheric studies. These observations are analysed to characterize the shallow liquid fogs occurring at Summit.

**2. Experimental Design**

Figure 1 shows a plan view of the configuration of the CIBS instruments on the tower at Summit. The tower was positioned approximately 500 m east of the main camp, and the ICECAPS facility was 150 m to the northeast (true). While we refer to the instrument heights as 10 m and 2 m, they were actually installed slightly higher and their heights varied with accumulation and scouring around the tower. Over time, accumulation dominates and thus the 10 m (2 m) instrument, which

was located at ~12(3) m in 2012, was closer to 11(2) m by June 2014. The instruments were positioned on the tower so that they pointed to the southwest, into the predominant wind direction. Data were screened for wind directions susceptible to flow distortion caused by the tower super structure, defined by a wedge 310° - 130° using independent observations of wind acquired at ~10 m by the NOAA Global Monitoring Division (GMD) approximately 1 km southwest of the tower. Data acquired when the tower was downwind of the station operations were also rejected (conservatively defined as a 45° wedge centred on 315°).

When winds were < 0.5 m s⁻¹, the data were retained unless they were from the direction of station. This procedure removed 33.1% of observations. Thus, the analysis discussed in this study represents 67% of the wind conditions that occur at Summit.

Valid observations for each FM100 require availability of all ancillary measurements (Fig. 2a) and also that the wind direction relative to the probe inlet horn was within an acceptable range (refer to Supplement). For this work, fog microphysics are presented only for data collected when the wind direction was within 50° of the inlet horn while radiative, meteorological

and occurrence data are presented for all previously described valid wind directions with respect to the tower. Fig. 2b shows both the operational uptime and effective uptime (or microphysical analysis uptime) for the FM100s, given these criteria. The effective uptime varied between about 10% and 80%, depending on the month. The 2 m FM100 was not operational until April





2013 due to mechanical problems. In general, uptime was greater than 50% during the summer, but was less than 50% in winter. Downtime was typically due to data dropouts associated with cold-soaked electronics.

Wind velocity and direction are necessary for processing the data acquired by the FM100s. At Summit, wind measurements were acquired from Metek USA-1 sonic anemometers that were installed alongside both FM100s. These data were acquired at ~20 Hz and were averaged to 1-minute. The anemometers were operated year-round and were heated during icing conditions to prevent riming and frosting of the sensors. Overall, the sonic anemometer at 2 m was operational 74% of the time and the sonic anemometer at 10 m was operational 79% of the time during the study period. Gaps in the data at both heights were filled using the station measurements. Though local wind measurements are preferred, this is justified because for wind directions in the range used for analysis, the station data were well correlated with sonic anemometers (at 10 m (2 m) $r^2$ = 0.94 (0.88) and 0.88 (0.78) for wind speed and direction, respectively). With the station data supplementing the sonic anemometers, the total availability for wind measurements was > 99% for the valid range of wind directions at both heights.

The FM100 is a single-particle light scattering spectrometer. The instruments were adapted for cold temperatures by limiting internal ventilation of electronics and adding external insulation around the instruments. Ambient air is drawn into a contraction horn inlet using pumps installed on the tower. The air flow within the instrument (the probe air speed, PAS), is measured continuously with a Pitot tube located in each probe's inlet tunnel, was approximately 15 m s⁻¹ and 7.5 m s⁻¹ at 2 m and 10 m, respectively. Sample air in the instrument is drawn past a narrow 658 nm laser. Hydrometeors that pass through the inlet scatter the beam and a portion of the forward-scattered light (between approximately 3° and 12°) is collected by a detector. An equivalent optical diameter is then derived for each particle from the voltage measured by the detector, which is calibrated to the scattering cross sections for liquid spheres. The detectable particle size range is 1-50 μm and individual detections are binned to provide size distribution measurements at 1 Hz. The data were averaged to 1-min temporal resolution. Thus, for the present work, the term particle size refers to measurements of individual hydrometeors averaged over measurements made 60 times each minute and should be interpreted as an optically-equivalent diameter.

There are two main uncertainties associated with the FM100 measurement (Spiegel et al., 2012). The first is sizing ambiguities arising from the non-monotonic Mie scattering function used to convert voltage measured at the detector to particle size. Following Pinnick and Auvermann (1979) and Dye and Baumgardner (1984), ambiguous sizing bins were identified and combined. The second set of uncertainties involve sampling losses in the aspiration and transmission of particles. The data were corrected for these biases following the recommendations of Spiegel et al. (2012). Details of this post-processing can be found in the Supplementary document.

Between November and March, the FM100 heaters were frequently unable to maintain continuously ice-free Pitot tubes, resulting in disruptions to the monitoring of the PAS. The problem affected the 2 m instrument more because it was located closer to the surface where the temperature is typically colder, and the problem persisted even after insulation was applied to the instrument case. Analysis of the data, in addition to measurements made by station technicians, confirmed that the pumps continued to operate normally during this time. In spring, the return of sunlight was found to provide sufficient heating to correct the problem, even when temperatures remained low. The affected data were recalculated using the mean PAS for





normal operating conditions, which is fairly stable, varying by approximately ± 5% (1σ). Note that the pumps may operate more efficiently in colder temperatures, potentially producing a small positive bias in estimated PAS.

Several ICECAPS data sets were also used to support the analysis of the probe data, including from an Atmospheric Emitted Radiance Interferometer (AERI), a Millimeter Cloud Radar (MMCR), a Microwave Radiometer (MWR), a

Micropulse Lidar (MPL), radiosondes, and a sodar acoustic sounder. The AERI is a self-calibrated infrared spectrometer (Knuteson et al., 2004a,b) that acquires spectra at sub-minute intervals from about 490 to 3000 cm⁻¹ (3-20 μm) with a spectral resolution of ~1 cm⁻¹. The spectra were post-processed using a principal components algorithm that reduces spectral noise (Antonelli et al., 2004; Turner et al. 2006). Quality control procedures removed 7.7% of the data due to instability of the reference sources, excessive noise, and iced optics (Fig. 2a). The AERI is used to determine the phase of particles measured

by the FM100s, as described in Sect. 3. The MMCR and MPL data were used to identify tropospheric clouds. The MMCR is a zenith-pointing 35 GHz Doppler radar (Moran et al. 1998) with high sensitivity to cloud particles and little attenuation through the typical clouds observed at Summit. It samples with ~4-sec temporal and 45 m vertical resolutions. It has been used previously to identify precipitation at Summit by Castellani et al. (2015) and Pettersen et al. (2018). The MPL is a 532 nm depolarization lidar with 5-sec temporal and 15 m vertical resolutions. The MPL data product used here is from a phase-

resolved cloud mask described by Edwards-Opperman et al. (2017). MWR data are used to retrieve liquid water path (LWP) during fog conditions using a physical retrieval algorithm (Turner et al. 2007). The sodar (Neff et al. 2008) is a 2100-Hz acoustic sounder that samples ~1-sec with a vertical resolution of 1 m.  More details on each of these instruments and data streams can be found in Shupe et al. (2013) and references therein.

Broadband radiometric fluxes (Shupe and Miller, 2016) were measured by Kipp and Zonen CG4 pyrgeometers

(longwave) and CM22 pyranometers (shortwave). The processing of these data is described by Miller et al. (2015; 2017). Cloud radiative forcing (CRF) is defined as the instantaneous effect of clouds on the radiative flux at the surface. CRF is calculated by subtracting a modelled clear-sky estimate, as described by Miller et al. (2015), from the measured radiative flux. The cloud radiative effect of the longwave component (LWCRE) is calculated in the same manner as CRF but only includes measured and modelled estimates from the downwelling longwave (e.g., Cox et al., 2015). Sensible and latent heat fluxes

(Shupe and Miller, 2016) were estimated via the bulk aerodynamic method and a two-level approach (10 m and 2 m), respectively (Miller et al., 2017).

## 3. Classification

The FM100 observations may be liquid fog, ice fog, blowing snow, or snow. Thus, it is desirable to classify the probe observations in order to subset the scenes containing liquid fogs. To do this, information about particle phase, precipitation

occurrence, the presence of elevated cloud layers, and the likelihood of blowing snow is needed.  This information comes from a blowing snow/precipitation mask derived using meteorological measurements and the MMCR, particle phase derived from the AERI infrared spectral radiances, and the MPL cloud mask.



The classification procedure consists of five steps: (1) scenes containing near-surface particles are separated from clear boundary layer scenes using a number concentration ($N_c$, cm$^{-3}$) threshold in the 10 m FM100, (2) elevated cloud layers are identified, followed by (3) occurrences of blowing snow and snow, and then (4) the phase of the particles is determined. Finally, (5), (1)-(4) are combined optimally. Next, we will describe the individual classifications.

A large proportion of the observations at Summit are of particle concentrations with low density (< 1 cm$^{-3}$), which include observations that are not traditionally classified as clouds, such as snow, light blowing snow, and transient particles, but also include forming and dissipating fogs. It is desirable to set a threshold low enough to capture these various conditions. Using an FM100, Spiegel et al. (2012) set a threshold $N_c > 10$ cm$^{-3}$ for similar purposes, which is low enough to observe snow (Braham 1990). Figure 3 shows frequency of occurrence of FM100 observations classified as containing any type of surface-

based cloud (red/black lines) as a function of $N_c$. The other lines in the figure represent classifications of cloud types that are discussed later. Particles of any type are measured between 80% and 90% of the time at 10 m, but there is relatively little sensitivity to thresholds of order 10$^{-3}$ cm$^{-3}$ or smaller. Due to the consistent volume size of the FM100, this is also roughly the lowest $N_c$ it can measure. Therefore, $N_c = 10^{-3}$ cm$^{-3}$ is a natural lower threshold for this study.

Events composed of ice are distinguished from those composed of liquid using the AERI data, which is collected at

intervals of about 20-75 second. These spectra are linearly interpolated to the regular 1-min sampling used for the FM100 data. The imaginary component of the complex index of refraction, which is proportional to absorption, has different spectral dependencies for ice and liquid in the infrared. Previous studies have exploited these dependencies to infer phase using spectral differencing techniques (Strabala et al., 1994; Turner et al., 2003). Particle size and habit also exhibit spectral dependencies, which is a large source of uncertainty in these methods, as is uncertainty in water vapour amount and cloud temperature. In

general, the uncertainty also increases for optically thin clouds because of reduced signal and for optically thick clouds because of loss of spectral structure (Turner, 2005). However, the spectral differencing approach is justified for this work because the cases of interest are less sensitive to the associated uncertainties. First, atmospheric transmission between the surface and the cloud is close to unity in the dry Arctic atmosphere (Turner et al., 2003) and can be assumed to be unity for the present purposes because the focus is on clouds with bases at the surface. Second, cloud temperature is well-characterized because it is measured

in situ. Finally, the scenes that are tested are single-layer, negating ambiguity from multiple cloud layers or multiple layers within a cloud.

Calculations of cloud emissivity are used for the phase identification using spectral microwindows that are sensitive to cloud detection (Turner et al., 2003). The necessary radiative transfer calculations were performed using the Line-by-line Radiative Transfer Model (LBLRTM), version 12.2 (Clough et al., 2005). Inputs to LBLRTM include twice daily radiosonde

profiles from Summit and estimates of trace gas profiles, as described by Cox et al. (2014). A small positive bias common in AERIs [less 0.5 mW m$^{-2}$ sr$^{-1}$ (cm$^{-1}$)$^{-1}$] was estimated empirically for each microwindow by comparison to the radiative transfer calculations during clear days and was subtracted from the microwindow radiance before analysis.

We use two spectral cloud emissivity differencing tests. The first, adapted from Strabala et al. (1994), is a threshold set to the 11 μm minus 12 μm emissivity versus the 11 μm minus 8.1 μm emissivity. The threshold to separate the clusters was



qualitatively set to $y = -2.5(\varepsilon_{11.12})$. As indicated by the example in Figure 4a, the clusters separate quite distinctly and therefore the precise slope of the threshold is not very important. The second test is applied in the far-infrared where ice and liquid absorption characteristics are very different and determines whether the 11 μm minus 17.8 μm emissivity is greater or less than zero (see example in Figure 4b). If both tests agree, the identification is deemed valid. If the tests disagree, the observation

is considered ambiguous. Generally, ambiguous identifications are expected when the microwindow differences are near 0, which are also likely to be cases when the absolute emissivity of the fog was quite low. However, Figure 4b also indicates that a small number of cases are not clustered as expected in the far-infrared, even when the microwindow signal differences are relatively large (red points near the centre of the figure). The reason for these ambiguous identifications is not clear and their source may be from instrumental errors or environmental conditions. These cases represent < 1% of the data, and the

democratic methodology has successfully screened them out.

      Scenes with elevated cloud layers are identified to determine the correspondence between the formation of hydrometeors near the surface and the presence of otherwise clear-skies, and because the phase classification described previously is only valid when no other clouds are present in the scene. Because no phase classification is possible, cases with elevated cloud layers are rejected from analysis of the fogs presented later. Using the MPL cloud mask, elevated clouds are

defined as clouds with bases above 200 m, similar to the definitions used by Starkweather (2004) and Castellani et al. (2015).

      Radar reflectivities greater than -5 dBZ, which are indicative of snow (Shupe et al. 2013), are used to identify precipitation falling between 200 and 300 m. These particles are assumed to be formed above the height of blowing snow and fall to the surface as precipitation and are therefore classified as snow. The radar data are combined with wind parameterizations for lofting snow calculated for western Canada by Li and Pomeroy (1997), which is classified as blowing

snow. When the radar data indicates snow and the lofting parameterization indicates blowing snow, the classification of a combination of snow and blowing snow is given.

      Figure 5 summarizes the fractional occurrence of the classified cloud types. Particles were observed in the lowest 10 m a majority of the time in all months, from 65% of the time in July to > 85% of the time in winter. Liquid fogs under otherwise clear skies were classified between 1% of the time (April) and 12% of the time (September); if liquid fogs could be reliably

detected in the presence of tropospheric clouds, these percentages would likely be higher. While most common in late summer, liquid fogs were also identifiable 3-7% of the time January-March.

      Recall from Figure 3 that the threshold for identification used to construct Figure 5 is small ($10^5$ cm$^3$). Figure 3 also shows that the threshold at which events begin to be missed, and the rate at which missed events increase, is different for each class. For example, as expected, low density types such as snow also require a low threshold to be captured while high density

types such as the fogs are relatively insensitive to the threshold.

### 3.1 Summary of classification

The classification scheme reveals several key features of hydrometeor occurrence over the Greenland ice sheet:



1. Precipitation occurred predominantly from July through October (similar to the study by Pettersen et al., 2018). This seasonal cycle is shifted later in the year compared to total precipitation amount, which peaks in July (Castellani et al., 2015).

2. Snow occurred more often without significant blowing snow, and blowing snow alone was most common in late winter and spring.

3. A large frequency of events (composite annual average = 27.3%) were detected by the FM100 under tropospheric clouds, which are common at Summit (Miller et al. 2015).

4. Identification of ice fog was less frequent than liquid and ice fog had a distinctly different seasonal cycle than that of liquid, with a peak in April (9.2%) and a low in July (0.3%).

While remarkable at first glance, the fact that there were fewer ice identifications than liquid does not necessarily indicate that in situ formation of ice near the surface at Summit is less common than liquid formation for three reasons: 1) the number density of liquid fog is expected to be higher and liquid more efficiently emits in the longwave, so therefore liquid is more likely to be identifiable using the AERI (i.e., it is associated with a larger infrared signal), 2) some ice fog events may be incorporated within the blowing snow or snow categories, and 3) the phase partitioning for events from which phase could not be retrieved using the AERI is unknown.

## 4. Liquid Fog Case Studies

We contrast two cases of liquid fogs to illustrate the conditions captured with the categorization scheme. We first discuss a case of near idealized fog formation conditions that appeared on the 16ᵗʰ of June 2013 in Sects. 4.1. Then, this case is contrasted with a winter case from January 2014 in Sect. 4.2.

### 4.1. Case of 16 June 2013

Clear skies persisted from 15-24 June 2013, and liquid fogs were observed in the early morning on most days during this period. Indeed, while this case represents an ideal illustration of fog formation at Summit, several similar cases can be found in the dataset. Figures 6a and 6b show the time-particle size cross-sections of $N_t$ from the FM100s beginning at 2000 UTC on 15 June extending through 1400 UTC on the 16th. During this period, the 10 m air temperature was -15 to -22 °C with light southerly winds (Fig. 6e). At 10 m, condensate first occurred ~0010 UTC as the solar elevation angle (SEA) dipped to 10° (< 20 W m⁻² SW_net, Fig. 6d) and 7 hours into development of the surface-based inversion (Fig. 6d). Initiation was followed by a period of uniform growth lasting 4-5 hours that closely aligns with a theoretical growth curve (Houghton 1985), calculated assuming a constant supersaturation of 0.1% (Fig. 6a). During the first hour of growth, the droplets reached approximately 25 μm in diameter after which the growth slowed and deviated from the theoretical curve, with particles eventually reaching ~40



μm. The curve neglects the gradual decrease in supersaturation that accompanies condensation without replacement of moisture, implying a decrease in supersaturation during development and indicating that the moisture source did not introduce new vapour as quickly as it was condensed. While most of the droplets closely followed the main growth curve, nucleation of new droplets continued until about 0300 UTC. The fog disappeared completely by ~1030 UTC when the sun reached an elevation angle of ~30°.

The sodar detects thermal turbulence where mixing occurs within a vertical temperature gradient. Note that structure in the sodar record may only reflect structure in temperature and not necessarily the boundaries of the fog, although in some cases radiative cooling at the top of the fog may account for the thermal contrast that is observable with the sodar. The sodar record shows a typical pattern of a "night time" stable surface layer 10-25 m deep before 0700 and after 2000 UTC (Figure 7a). This surface layer represents a shallow, but strong temperature inversion embedded within a deeper inversion that extended about 100 m above the surface at initiation, deepening to several hundred meters 12 hours later. The surface layer increased in depth from about 15 m to about 65 m during the course of the duration of the fog layer, transitioning from statically-stable to a shallow convective layer in association with the diurnal cycle (Figure 7a). The convective plumes are visible in Fig. 7a as vertically oriented echoes below the red dashed line with intervals of order minutes. Dissipation occurred shortly after the onset of convection ~0845 UTC (SEA ~23°, $SW_{net} = 61$ W m$^{-2}$, SWD = 426 W m$^{-2}$), and the fog disappeared when the convection reached a developed stage around 1000 UTC (SEA ~29°, $SW_{net} = 87$ W m$^{-2}$, SWD = 562 W m$^{-2}$).

Particles were observed at 2 m 10-20 minutes after initiation at 10 m. Both the $N_c$ and the width of the size distribution were larger at 2 m than 10 m, consistent with particle formation near 10 m followed by settling and evaporation (~0.01 m s$^{-1}$). At Summit, condensation in radiation-induced fog frequently occurs near 10 m because non-linearity between temperature and saturation within the inversion leads to supersaturation first between 3 and 18 m, with lower vapour pressures both above and below this layer (Berkelhammer et al., 2016). A lag in $N_c$ of 3 to 4 min between 10 m and 2 m near the peak particle size is evident in the time series (Fig. 6c), implying a maximum settling rate of 0.03 to 0.04 m s$^{-1}$.

Between 0200 and 0500 UTC, the 2 m air temperature deviated by ~ +5 C from the smooth diurnal cycle that is evident both before and after the fog, and the LWP increased to ~ 15 g m$^{-2}$ with no clouds observable above the fog layer by the radar or lidar. This corresponded to an increase in CRF from < 10 W m$^{-2}$ to ~65 W m$^{-2}$, caused primarily by increased downwelling longwave, as evidenced by the LWCRE (Fig. 6f). These values are within the range of the thickest clouds at Summit (Cox et al. 2014; Miller et al. 2015) and large enough to drive latent and sensible heat fluxes to near-neutral (Fig. 6g), indicating a well-mixed surface layer. As we will see later, fogs generally produce closer to ~ 10-20 W m$^{-2}$, but larger values such as the 16 June case occur occasionally. Because of these factors, the temperature inversion within the surface layer completely eroded by around 0330 UTC. Surprisingly, this did not dissipate the fog, and, in fact, an increase in $N_c$ at 2 m occurred in conjunction with a brief interruption in the otherwise smooth growth rate. Following the brief interruption, the particle size and $N_c$ were larger than before with $N_c$ at 2 m exceeding 200 cm$^{-3}$; as we will see later, such high $N_c$ as observed from 0400-0500 UTC are atypical.





Thus, while the fog was likely induced by radiation initially, it was maintained, and ultimately continued to grow without additional infrared loss at the surface driving saturation in the air column. Indeed, with warming of the surface layer, introduction of large quantities of vapour would have been necessary to maintain the fog. While the relatively large LWP (Fig. 6f) implies significant cloud-top radiative cooling (and thus, plausibly buoyancy-driven mixing) may have occurred. The

necessary moisture was thus more likely introduced to the surface layer from above by mixing generated by wind shear. The sodar data shows signatures of turbulence in the form of Kelvin-Helmholtz instabilities (Figure 7b) driven by shear at the top if the surface layer (30-50m) between 0300 and 0500 UTC, corresponding to the period of time with enhanced LWP, CRF, and $N$, and generally more variable particle size at both measurement heights. Indeed, in June, the firn is generally colder than the surface (Miller et al., 2017) so vapour transfer tends to be downward, toward the surface, from the atmosphere because the

mixing ratio is higher in the (saturated) air immediately above the surface than in the firn (Berkelhammer et al., 2016). The latent heat flux (LH) was positive (defined positive into the surface) for the duration of the case study, consistent with the hypothesis that the fog was driven by moisture from aloft (Fig. 6g). Therefore, the moisture source for this case was likely the atmosphere and not the local surface. Interestingly, the $N$ at 10 m was consistently lower than at 2 m. The reasons for this are unclear, but since there were also more smaller particles at 2 m, it is possible that concentration of particles at 2 m associated

with partial evaporation and subsequent slowing of their descent was a factor. The difference may also be associated with the heights of the instruments relative to the height of maximum supersaturation (see Fig. 2 by Berkelhammer et al. (2016)).

### 4.2. 16 January 2014

Figure 8 is similar to Figure 6, but for a wintertime case. The fog was detected at ~0530 UTC at 10 m, and the particles grew quickly to ~35 μm within approximately 30 minutes (Figs. 8a,b). The $N$ at both heights was about an order of magnitude

smaller than the June case (Fig. 8c). The troposphere was clear with intermittent ice clouds above 2 km (Fig. 8d). Similar to the June case, the wind was southerly and light with lower wind speeds during the period of the fog (Fig. 8f). The net atmospheric heat flux was generally negative (Fig. 8h) due to radiative cooling at the surface that maintained a strong inversion with a gradient of ~ 10 °C between 2 m and 10 m (Fig. 8e). There was not enough LWP for reliable detection by the MWR (uncertainty ~5.5 g m⁻³), but the CRF was between 10 and 40 W m⁻²; note that the upper level clouds contributed some to this

forcing, in particular after 1430 UTC. The CRF quickly spiked to 40 W m⁻² during the initial fog development around 0600 UTC, which was also the time period when the $N$ was the highest. Later increases in CRF also corresponded in time to increased $N$. Consistent with variable wind direction, the apparent intermittence of $N$ may be explained by a spatially heterogeneous fog periodically passing the tower instruments.

This case was likely a mixed-phase fog with a liquid formation layer precipitating into an underlying settling layer

composed of homogeneously-frozen ice. The air temperature at 10 m (near the formation height) was between -35 and -38 °C. While the homogeneous freezing point for liquid is imprecise and dependent on conditions (generally about -40 °C), the 2 m air temperature and skin temperatures were ~ -47 °C and -50 °C, respectively, both too cold to support liquid particles. Thus,



it is likely that the particles observed at 2 m were frozen droplets. While we can only infer the phase of the underlying layer from the measured temperature, we do have confidence that the droplets that formed aloft were indeed liquid: The AERI classifications were not ambiguous as 176 of 177 samples for this event were identified as liquid; the AERI classification should not have been confounded by the presence of the ice because the viewport for the AERI is a few meters above the

surface, and therefore near the top or above the ice layer; the AERI results are also supported by the MPL data, which indicates mostly liquid at levels where the MPL has sensitivity, between 100 and 150 m (Fig. 8d).

The ceilometer (not shown) and MPL (Fig. 8d) data indicate a deeper fog layer than the previous case, which may have been enabled by generally deeper and more persistent inversions in winter (Miller et al., 2013); the height of the maximum temperature within the troposphere for this case was 300-400 m compared to ~ 100 m at initiation for the June case. However,

the sodar record (Figure 7b) shows that the depth of the surface layer embedded within the deeper inversion for the winter case was actually much shallower than in the summer case, < 10 m during the duration of the fog. The fog that is within the surface layer is the fog that was sampled by the measurements made at the tower (e.g., Figs. 8a,b) and may be distinct from the deeper fog layer visible in the MPL data: The sodar record also shows a significant amount of structure throughout the upper fog layer indicating additional stable layers (Fig. 7c). This structure is physically decoupled from the surface layer, separated by a thin

layer of low reflectivity that may be indicative of a very low-level jet and likely suppresses mixing between the layers. However, the layers may be dynamically coupled in other ways. Buoyancy waves with periods of a few minutes to 15 min in the upper layer have an observable remote influence on the surface layer through fluctuations in the horizontal pressure field that produce a moving pattern of convergence and divergence (not shown). It is unknown if the observed microphysics below 10 m are representative of the fog above the surface layer. However, radiatively, the combined physically-thick layer of fog

compensates somewhat for the low $N$. observed near the surface, enhancing the CRF relative to the summer case with a larger optical depth. The strength of the inversion also contributed to enhancing the CRF because the particles were considerably warmer than the surface.

Interestingly, the fog development coincides closely with the weak early-season diurnal cycle. In mid-January, the sun does not get above the horizon at Summit but is about 3° below the horizon at solar noon (Fig. 8e). Atmospheric scattering

in these twilight conditions produced 1-3 W m² of diffuse incident solar radiation (Fig. 8e) that corresponded in time with the fog initiation. Though the mechanism is unknown, there is a possibility that the fog was diurnally forced, similar to the June case, yet unlike summer, the timing was coincident with peak in solar radiation rather than the minimum.

The stably-stratified surface layer was largely isolated from the free troposphere in this case. The temperature gradient within the firn (which is warmer at depth than at the surface) produces a constant supply of vapour towards the surface from

below, providing moisture for the fog, which is then returned via settling (Berkelhammer et al., 2016). As before, we observed higher concentrations and more smaller particles at 2 m than 10 m.



## 5. Statistics of Fog Properties, 2012-2014

### 5.1. Physical Properties

Figure 9 shows distributions of the particle sizes for liquid fogs (blue) compared to those for the ice categories; the figure displays sensor height as rows (10-m – top row, 2-m - bottom row) and season as columns (left column - low-light winter season (NDJF) and right column - sunlit summer season (JJAS)). Ice particles are non-spherical and, thus, the distributions represent an effective size with reference to the scattering properties of spherical liquid; asphericity and orientation are important factors in the sizing of ice using a scattering spectrometer that impose significant uncertainties (Borrman et al., 2000). Thus, the distributions of ice classes should be treated cautiously and are shown here for context. The liquid classification stands out distinctly from the ice both in the shape of the distribution and the overall $N_c$ within each size bin. For smaller particles, liquid particles are present in higher concentrations than ice particles in summer and are smaller in winter, while the opposite is generally true for larger particles. Despite the uncertainties in sizing ice particles, the relative $N_c$ within each size bin separates logically between the different types. For example, there are more particles in blowing snow at 2 m than at 10 m, while the two heights show similar concentrations of ice fog. Also, the occurrences of snow have consistently low $N_c$.

As implied by the case studies, fogs in winter have lower $N_c$ overall compared to summer. This relationship between temperature and fog $N_c$ is evident when analysed directly: $N_c$ is smaller in colder fogs, with temperature being correlated with the log of $N_c$ (r=0.54). Gultepe et al. (2002; 2004) reported a similar finding in tropospheric Arctic clouds containing super-cooled liquid. Note that while aerosol concentration is likely a factor, dynamical and thermodynamical processes can also play a role (Gultepe et al., 2002; Gultepe and Isaac, 2004).

The distribution of liquid particles in summer peaks between 20 and 25 μm in diameter at 2 m (Figure 9b) while the distribution at 10 m (Figure 9a) is broader with more small particle sizes. This is consistent with particles preferentially forming near 10 m before settling to 2 m. Thus, multiple growth stages are likely represented in the distribution at 10 m, while the distribution at 2 m is more idealized because it is composed primarily of mature particles. Though the winter liquid distributions are more difficult to interpret owing to a limited amount of data, this finding is supported by a calculation of the effective diameter (Hansen and Travis 1974) for all liquid fog scenes in all months, which shows a correlative relationship between effective diameter and $N_c$ at 10 m (Figure 10a) (r = -0.55), but not at 2 m (Figure 10b). This indicates that at 10 m, when the $N_c$ is high, the particles are small (forming), while at the same time low $N_c$ for large particles is consistent with loss via settling out of the layer. While a similar relationship might be expected from a range of aerosol concentrations, if aerosols were the explanation for the observations at 10 m, a similar result should be found at 2 m, but it is not.

When considering the overall distributions of $N_c$ (Figure 11), at both heights the concentrations in winter are smaller than in summer. In winter, there are typically more particles at 2 m than 10 m whereas during summer the differences between





the heights are less discernible. The median value of $N_c$ was typically < 7 cm$^{-3}$ in winter and between 5 and 20 cm$^{-3}$ in summer (Fig. 11).

Figure 12 shows the temperature distributions for times classified as liquid fog and ice fog (the two types associated with in situ formation). In Fig. 12, $N_c$ was only used for identification of events. Thus, precise estimates of $N_c$ are not important,

and the threshold for wind direction is less restrictive than the microphysical results shown in Figs. 9-11. Consequently, larger sample sizes are incorporated into this analysis. Three different temperatures are plotted beginning with brightness temperatures derived from near-saturated $CO_2$ emission between 675-680 cm$^{-1}$ measured by the AERI in Fig. 12a. These frequencies are sensitive to the lowest few 10s of meters above the instrument and may be most similar to the fog thermodynamic temperature. Since liquid freezes homogeneously near -40 °C, observations at lower temperatures are not

possible; less than 1% of observations classified as liquid occur at temperatures < -40 °C in Fig. 12a. The air temperatures at 10 m (Fig. 12b) are quite similar to the AERI-derived temperatures. Air temperatures closer to the surface at 2 m (Fig. 12c) are generally colder and include many more instances with temperatures below -40 °C. Note that this is not an indication of water existing in a liquid phase below -40 °C, but rather indicates that the liquid layer lies above this cold air where the temperature is warm enough to permit the existence of liquid droplets. Between November and March, two thirds of the liquid

fog identifications occurred when the 2 m air temperature was below -40 °C. This result indicates that the 16 January case is typical. A stretch of such occurrences was observed when the 2 m air temperature was < -45 °C during an extended clear-sky period during the last two weeks of March 2013. Unlike the 16 January case, for cases during the spring and autumn, there was sufficient daylight for images acquired by cameras mounted to the tower to show evidence of optical phenomena caused by liquid droplets. Images of fogbows appeared at times that coincided with the identification of liquid fogs in March 2013.

For both ice and liquid identifications in Fig. 12a-c, the temperatures at which they occur are lower in winter and higher in summer. Liquid generally occurs during cooler temperatures than ice in summer and thus the seasonal cycle for liquid is muted compared to that of ice. This is likely a result of diurnally-forced radiation fogs occurring during the colder part of the day in summer. There is also a notable difference in the timing of the seasonal cycle with the liquid tending to be warmer than ice in the spring transition and cooler than ice in fall. It is unclear whether this is more closely tied to the seasonal cycle

in aerosols (e.g., Schmeisser et al. 2018) or to meteorology. The wind regimes are similar between ice and liquid in winter, but liquid fogs in summer occur during particularly calm conditions in association with the diurnal development of stable stratification (Fig. 12d).

### 5.2. Radiative Properties

The impact on the surface radiation budget from the various classifications is evaluated in Figure 13. The sample

sizes in Fig. 13 are analogous to those in Fig. 12. First, distributions of LWCRE for the classes appear alongside that for all observations in Fig. 13a. The peak for both liquid fog and ice fog are close to 10 W m$^{-2}$. Both ice and liquid show long tails to larger values and thus the mean is 19.8 (median =14.8) W m$^{-2}$ for ice and 26.1 (median = 18.1) W m$^{-2}$ for liquid. It is notable





that unlike distributions of temperature and microphysics shown earlier, the distributions of LWCRE for ice and liquid are similar. It is therefore possible that a similar amount of downward longwave radiation is needed, when both liquid and ice fogs form, to achieve radiation balance between the surface and the lower atmosphere. However, note that the sample selection was radiative in origin in the first place, and the proportions of missed classifications, in particular at the low end of LWCRE, may

be different for the two types. The other ice classes, snow, blowing snow and a mixture of both are also plotted in the panel for context. Blowing snow is distributed fairly evenly across the range of LWCRE with the larger values more likely to be associated with higher wind speeds ($r = 0.31$). Snow also is distributed over a wide range of values, but generally higher than blowing snow alone, which is expected because LWCRE during snowy conditions is associated with a precipitating cloud whereas blowing snow may occur under otherwise clear skies. When snow and blowing snow occur together, the distribution

is clustered near the largest values, a condition mostly associated with storms.

When the total CRF is considered (Fig. 13b), the results are similar. The forcing is smaller because of the addition of shortwave cloud-cooling, but only slightly smaller because the high year-round albedo at Summit limits the ability of clouds to cool the surface there (Miller et al., 2015). Notably, there is increased separation in CRF compared to LWCRE between the ice fog class ($\mu = 12.1$ W m$^2$, median = 7.9 W m$^2$) and liquid fog ($\mu = 22.1$ W m$^2$, median = 17.1 W m$^2$). This is because a

larger proportion of ice fog occurs at higher sun angles when the shortwave cloud-cooling effect is larger. Therefore, the time of day when liquid fogs typically occur maximizes their net radiative forcing.

The lower panels in Figure 13 give the statistics for each month and annually for liquid (Fig. 13c) and ice (Fig. 13d). The annual mean CRF for liquid fogs under otherwise clear skies is 1.5 W m$^2$ when normalized by the frequency of occurrence. The maximum occurs in July (2.6 W m$^2$) and the minimum in April (~ 0 W m$^2$). Due to subsampling, this estimate is probably

slightly lower than the total annual CRF from the fogs under all sky types because fogs that were present under optically-thin cloud cover may have contributed additional CRF. However, events that were missed for being too thin contributed very little radiatively (and consequently, no identification was possible), only a very small number of "too thick" (0.8%) were identified, limiting the influence of this class on the mean CRF, and when ambiguous identifications are included in the analysis (not shown), the annual mean CRF decreases slightly to 1.2 W m$^2$. For ice, the normalized annual mean is 0.7 W m$^2$. While the

magnitude of the values is similar between ice and liquid, the seasonal cycles are different, with ice peaking in July/September (2.2 W m$^2$). For reference, the mean annual CRF for all sky conditions at Summit is 33 W m$^2$ and is positive in all months (Miller et al., 2015).

## 6. Discussion and Conclusions

This study analysed in situ measurements of fog properties over the Greenland Ice Sheet at Summit Station within

the predominant southerly-to-westerly wind regimes. At 10 m above the surface, particles associated with snow, blowing snow, fog or transient particles were observed more than 60% of the time in all months and over 90% of the time in winter. Liquid fogs occurring under otherwise clear skies are observed in all months, peaking in September and with a minimum in April.



Generally, winter fogs had significantly lower number concentration ($N_c$), but only slightly smaller particle sizes compared to summer when mature particles were measured to be typically 20-25 μm in diameter, similar to reports from Borys et al. (1992). Typically, we find that $N_c$ in fogs are < 7 cm$^3$ in winter and between 5 and 20 cm$^3$ in summer. While these are lower number concentrations than the 30 to 165 cm$^3$ reported by Borys et al., their study was limited to six well-developed fog cases in

August. Their results are actually similar to the well-developed fog analysed as a case study discussed here in Sect. 4 when $N_c$ was briefly observed above 200 cm$^3$.

In isolation, the average cloud radiative forcing (CRF) from the liquid fogs when the sky was otherwise clear was 26.1 W m$^2$ but was much higher in some cases. When normalized by the fractional occurrence when no other clouds were present, the annual average CRF for liquid fog was 1.5 W m$^2$. As discussed, this estimate may be conservative given sub-

sampling. Given the relatively small overall forcing, it may be more instructive to consider the role of fogs at times in which they occur. Fogs that occur in summer generally appear near the cold times of the diurnal cycle when the surface layer is typically stably-stratified. For the 16 June 2013 case, the fog produced enough radiative forcing to increase temperatures by about 5 °C. Thus, the development of fog under these conditions constitutes a negative feedback on surface temperature during the coldest part of the day. While surface melt is rare at Summit, at lower elevations where melting produces runoff and is

more frequent, this damping of surface cooling by fogs could precondition the surface for melting later in the day.

Previous analysis of vapour isotope profiles up to 38 m at Summit (Berkelhammer et al., 2016) indicate that condensation occurs preferentially between 2 m and 10 m. In situ observations of particles from the FM100 scattering spectrometers at the two heights support this finding, showing distinct signatures of droplet growth near 10 m and concentration of mature particles near 2 m. The higher number concentrations observed at 2 m may indicate that surface riming is an

inefficient process, possibly associated with evaporation of the descending particles. Though most of the droplets nucleated at initiation of the fog in the case studies, some new droplets nucleated later. While recycling of aerosols following droplet evaporation is plausible, Bergin et al. (1995) found that large aerosols (> 0.5 μm) occurring in low concentrations were scavenged while populations of smaller aerosols (> 0.01 μm) were only partially activated.

The liquid fogs at Summit are super-cooled because temperatures are (nearly) always below freezing. However,

winter temperatures are frequently near or below -40 °C, which is approximately the homogeneous freezing point of liquid. We have observed liquid fogs to develop very close to this threshold, which can only occur in environments that are devoid of ice forming nuclei. Elevated fog formation at Summit has the important implication that it extends the season under which liquid fogs can form to the winter months. Two out of three scenes containing liquid fogs from November-March occurred when the surface was colder than -40 °C with skin temperatures as low as -57 °C. Particles were large enough to settle out and

were observed at 2 m for these cases. We postulate that such fogs were therefore mixed-phase, with a liquid formation layer residing above and feeding a settling layer of frozen ice particles. The resulting surface accumulation may be more likely to behave like light precipitation than rime with respect to surface roughness and has more potential to be re-lofted. Additionally, the relatively lower density of the ice particles compared to their liquid state could serve to reduce their setting rate, while the





phase change may buffer the settling particles somewhat from re-vaporizing, or even reverse the process, as their sublimation rate as ice would be weaker than their evaporation rate as liquid.

The intricate interplay between thermal, dynamical, and microphysical processes, coupled with the significant radiative impact of fogs, highlight that more work is needed to understand the dynamics of the boundary layer during fog
events. For example, the multiple fog layers apparent from the sodar record in the wintertime case study are intriguing, but will require additional analyses and possibly new measurements in order to ascertain the processes involved in developing and maintaining the distinct layers, as well as to identify the ways they are coupled and what role they may have in regulating the surface mass balance. A complete physical characterization of the fogs also requires detailed observations of aerosols, which were not collected during the period the FM100s operated. While aerosol optical properties are routinely observed at Summit
(Schmeisser et al., 2018), additional observations previously only made for brief periods (e.g., Bergin et al., 1995) of number concentration and speciation are necessary for further study. Indeed, such measurements are warranted as the influence of the fogs on climate is likely important for surface melt potential (this work), aerosol cycling (Bergin et al., 1995) and sublimation/deposition processes (Berkelhammer et al., 2016). We anticipate that these processes will act differently at other locations over the Greenland Ice Sheet where different boundary-layer characteristics occur, including wind regimes associated
with sloped topography (e.g., katabatic wind), cloud occurrence (e.g., Starkweather, 2004; Cox et al., 2014) and moisture availability. The findings presented here suggest that fogs significantly influence the surface mass and energy budgets over the Greenland Ice Sheet and therefore requires consideration when modelling ice sheet boundary-layer processes.

**Author Contributions**

DC, MB, VW and KS lead collected the observations with assistance from NM, MB and CC. CC lead the analysis with
contributions from DC, MB, WN, NM and MS. CC prepared the manuscript with contributions from DC, MB, MS, NM, and VW.

**Acknowledgments**

This research was supported by National Science Foundation grants PLR1023574, 1303879, 1314156, 1414314, and 1420932. CJC also received support from the Arctic Research Program of the National Oceanic and Atmospheric Administration
(NOAA) Climate Program Office (CPO). The broadband radiation data was collected by the Swiss Federal Institute, ETH; the data from Miller et al. (2015; 2017) is archived at the NSF Arctic Data Center, doi:10.18739/A2Z37J. Meteorological data collected by NOAA's Global Monitoring Division (GMD) may be accessed from https://www.esrl.noaa.gov/gmd/. The ICECAPS data is available from the NSF Arctic Data Center from the following dois: ceilometer (10.18739/A2221V), radar (10.18739/A2BJ3X, 10.18739/A2318G, 10.18739/A2121J), sodar (10.18739/A21V2V), radiosondes (10.18739/A2X508,
10.18739/A2NN44,   10.18739/A2WZ18),   AERI   (10.18739/A2TF7R,   10.18739/A2VF6P,   10.18739/A2JZ2J,



10.18739/A29F73, 10.18739/A2PJ65), MWR (10.18739/A22J6K, 10.18739/A2HJ57), and MPL (10.18739/A20R48, 10.18739/A2MJ55, 10.18739/A2V210). The cloud probe data and sonic anemometer data are available from https://www.esrl.noaa.gov/psd/arctic/observatories/summit/index.html. The Department of Energy's Atmospheric Radiation Measurement (ARM) Program provided the MPL and ceilometer. We appreciate the CIBS program data management and

field efforts of Michael O'Neill (formerly NOAA) and David Schneider (CIRES, NCAR). We acknowledge useful conversations on aerosols with Jessie Creamean (CIRES/NOAA) and instrumentation with Bill Dawson and Matt Freer at Droplet Measurement Technologies (DMT). We appreciate the efforts of David Turner (NOAA), Claire Pettersen (SSEC), Aronne Merrelli (SSEC), and Jonathan Edwards-Opperman (Univ. Oklahoma) in product development for the MWR and MPL data sets. We also appreciate the efforts of the technicians at Summit Station and Polar Field Services who provide high quality

science support as well as the efforts of the many contributors to the ICECAPS and CIBS research programs.

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





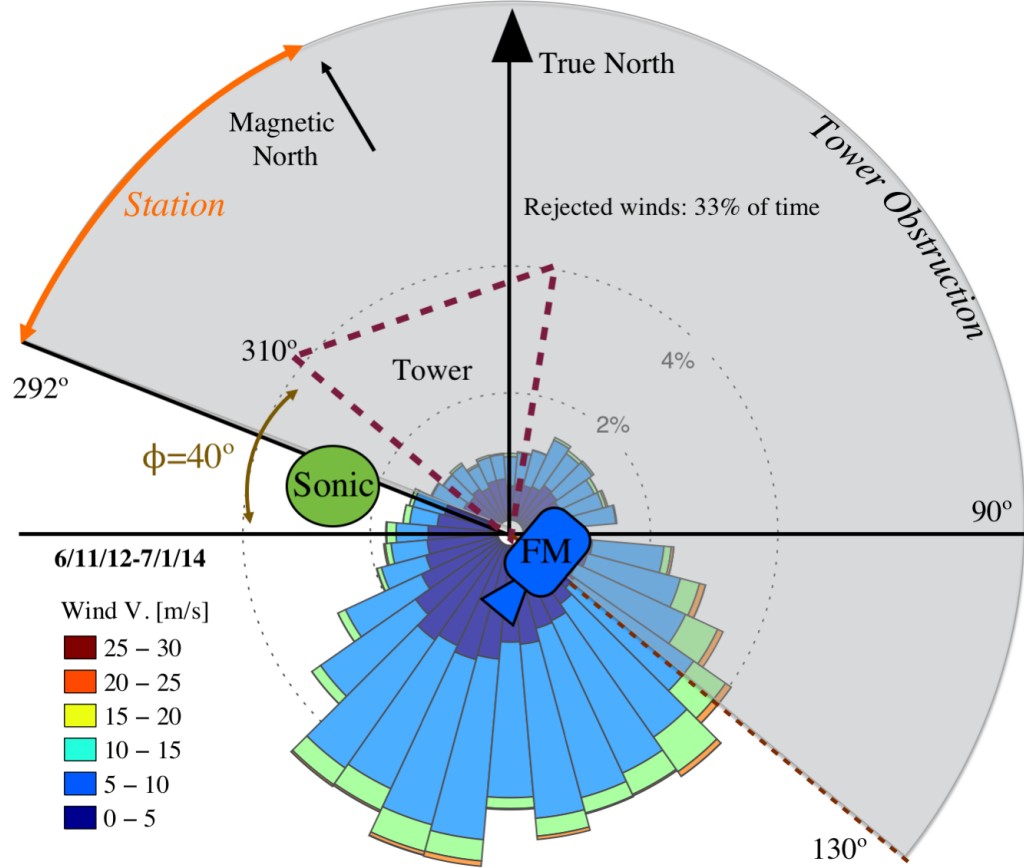

**Figure 1.** Schematic plan view of the field setup of the FM100 (blue icon labelled "FM") on the tower (dashed triangle) at Summit Station. The green oval ("sonic") is the position of the Metek sonic anemometer. The grey shaded area are the wind directions that were rejected and the orange arrow denotes the sector facing the station. The wind rose is also shown cantered on the FM position.





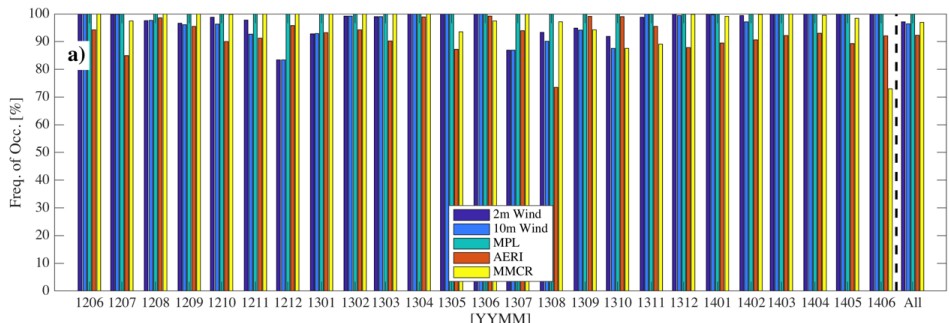

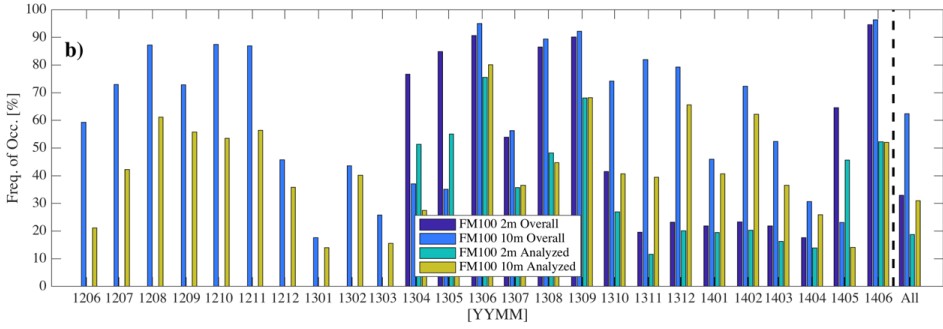

**Figure 2.** (a) Percent of available data: wind (blues), MPL (green), AERI (red) and MMCR (yellow) for each month in the study period. (b) Similar to (a), but for the FM100s at 10 m (dark blue/green) and 2 m (light blue/yellow). Blues show the amount of available data and yellow/green shows the amount of analysed data after screening for wind direction and availability of ancillary measurements from (a).





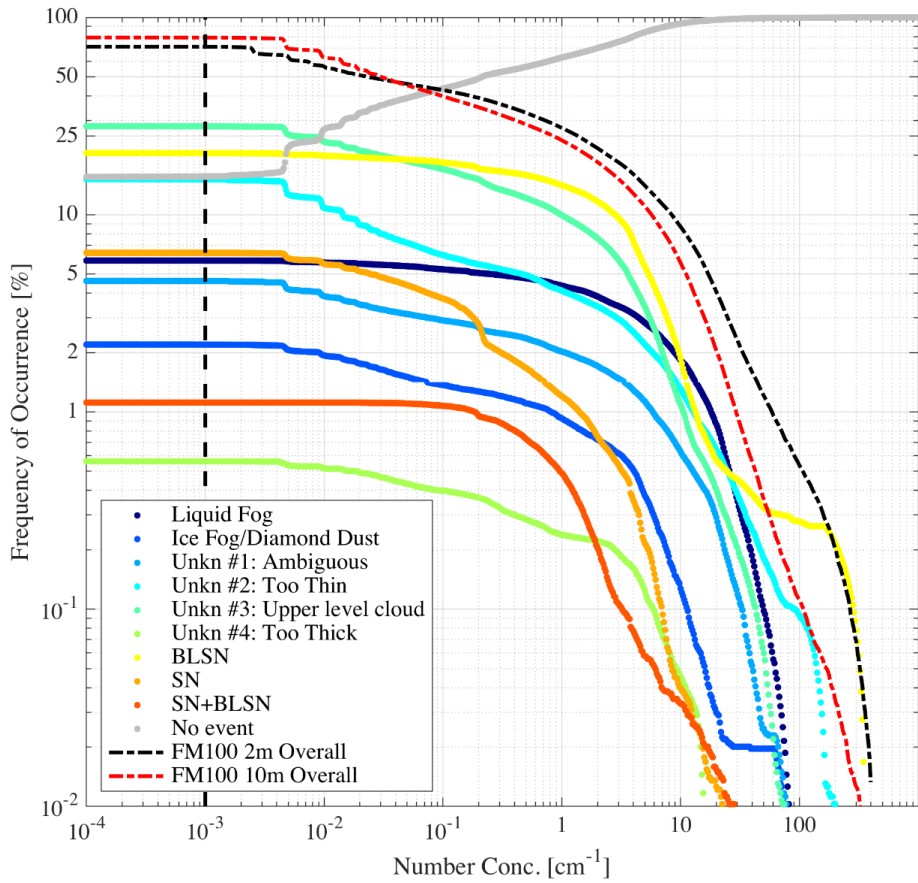

**Figure 3.** Percent of time clouds are identified in FM100 data at 2 m (black dashed) and 10 m (red dashed) as a function of threshold in number concentration. Colours show the same for the classifications that are reported in Figure 5.



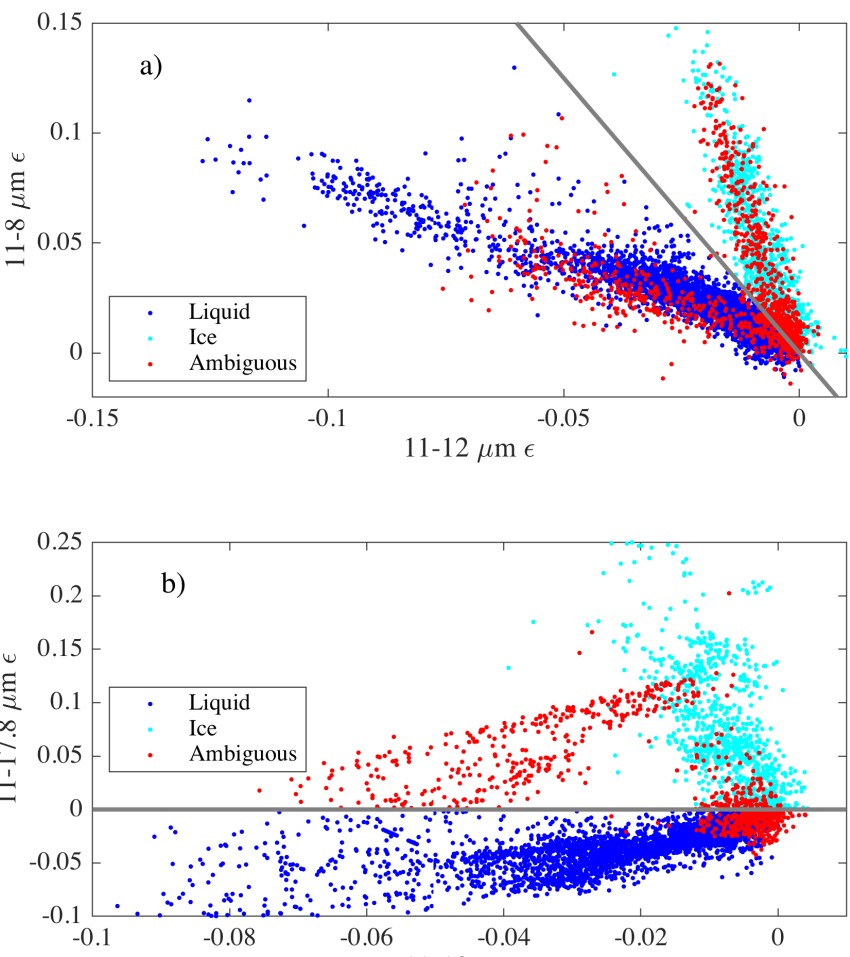

**Figure 4.** Example of phase classification using microwindow cloud emissivity measured by the AERI for the month of June.

5   (a) shows the first test described in the text and (b) shows the second. The colours indicate the classification. Only scenes
containing identified events where the AERI 11 µm emissivity was > 0.02 and no upper level clouds were detected are shown
(n = 4527).





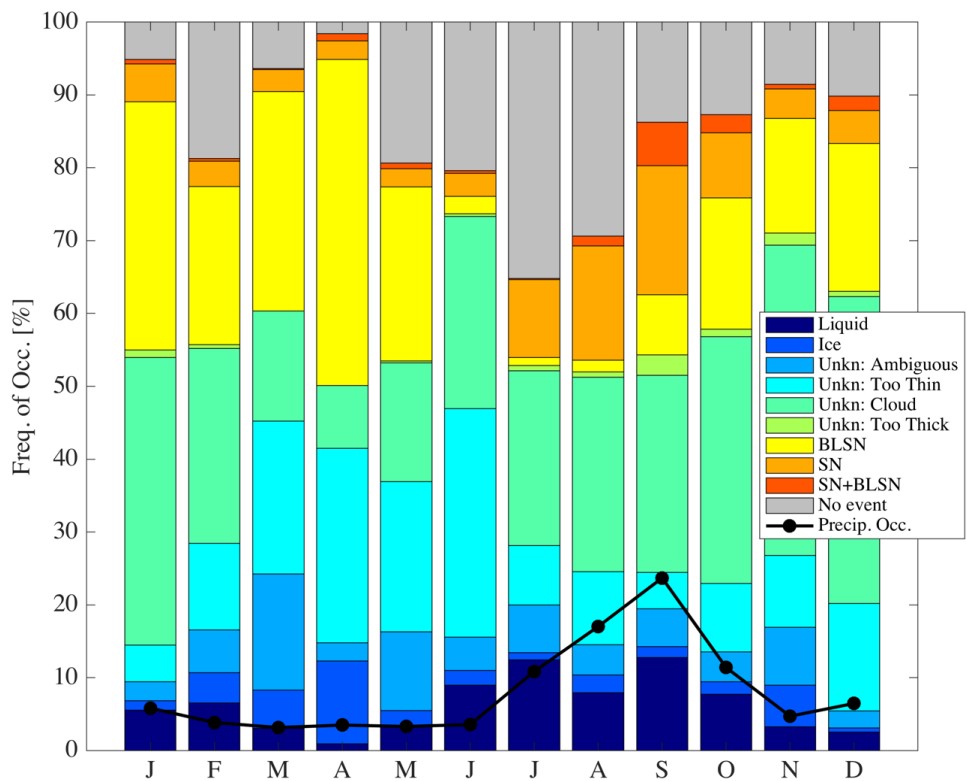

5 **Figure 5.** Composite monthly frequencies of occurrence of each classification. BLSN and SN refer to blowing snow and snow, respectively. Precipitation occurrence is the sum of SN and BLSN+SN.





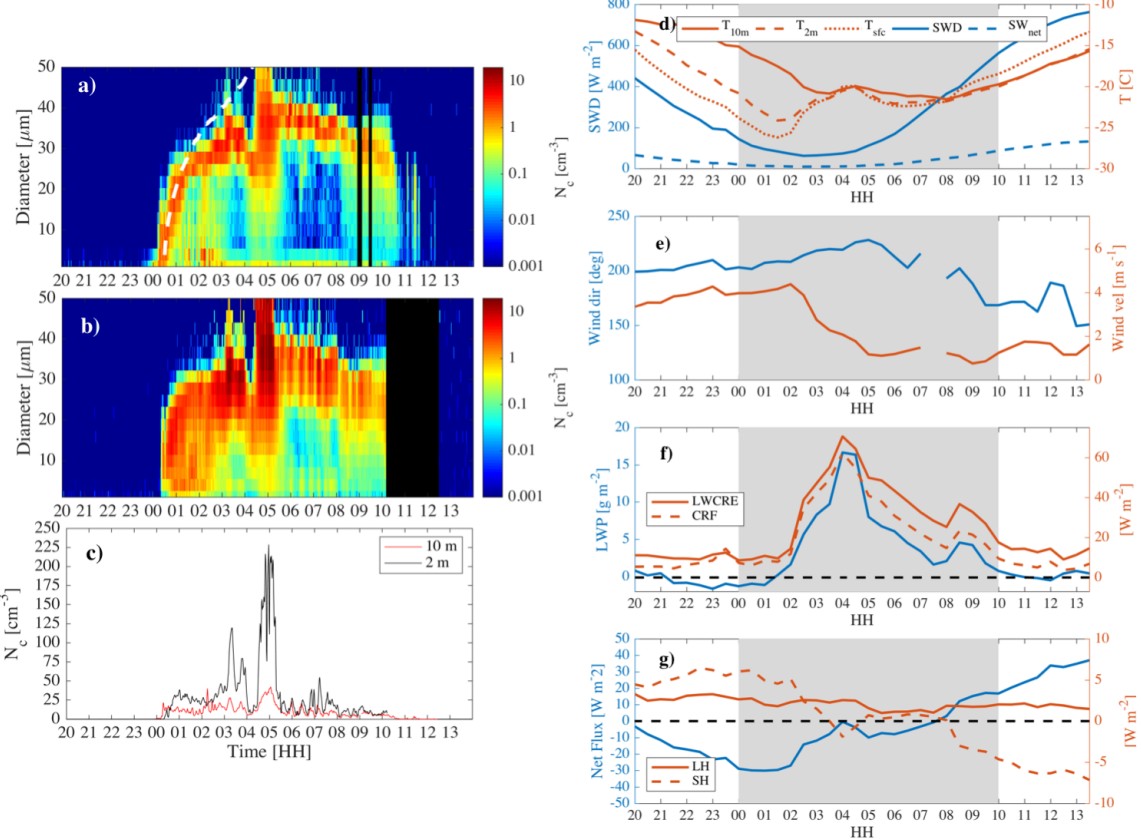

**Figure 6.** 16 June 2013 case study. Number concentration and particle diameter from FM100 at (a) 10 m and (b) 2 m. (c) total number concentration ($N_c$) in all bins for the 10 m FM100 (red) and the 2 m FM100 (black). (d) temperatures at surface (skin),

5    2 m and 10 m (reds) and downwelling shortwave radiation (SWD) and net shortwave radiation ($SW_{net}$) (blues). (e) wind direction (blue) and velocity (red). (f) longwave cloud radiative effect (LWCRE) and total cloud radiative forcing (CRF) (reds) and liquid water path (LWP) (blue). (g) latent (LH) and sensible (SH) (reds) turbulent fluxes and net flux (blue). The white dashed line in (a) is a theoretical growth curve calculated from Houghton (1985).





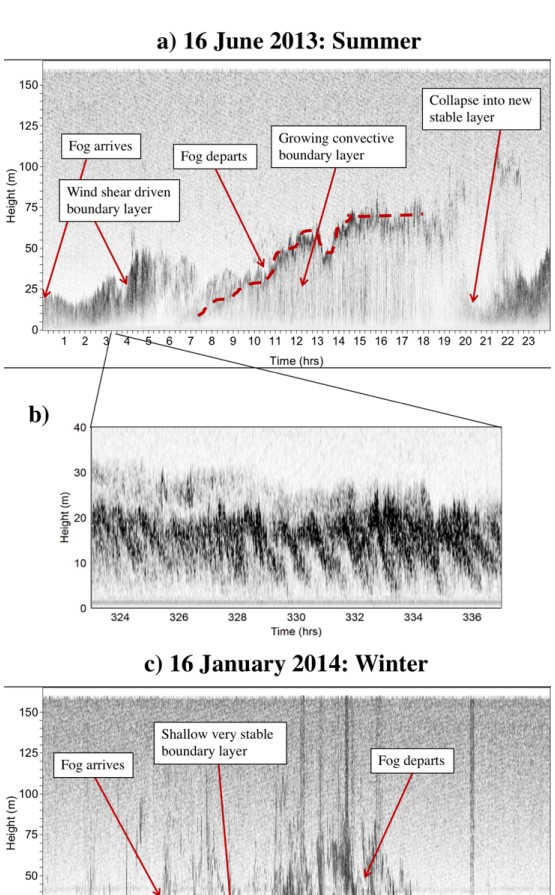

**Figure 7.** Sodar facsimile records between the surface and 160 m height for the summer case (16 June 2013) (a). (b) shows an expansion of a brief period from 0323-0337 UTC and 0-40 m height from (a) to highlight the Kelvin-Helmholtz instabilities observed at this time. (c) is similar to (a) but for the winter case on 16 January 2014. Dark features that extend to the top of the plot in (c) (e.g., near 2000 UTC) are likely noise from station activities and the horizontal feature in (c) near 40 m height is due to reflections of a side lobe from an object nearby on the ice surface.



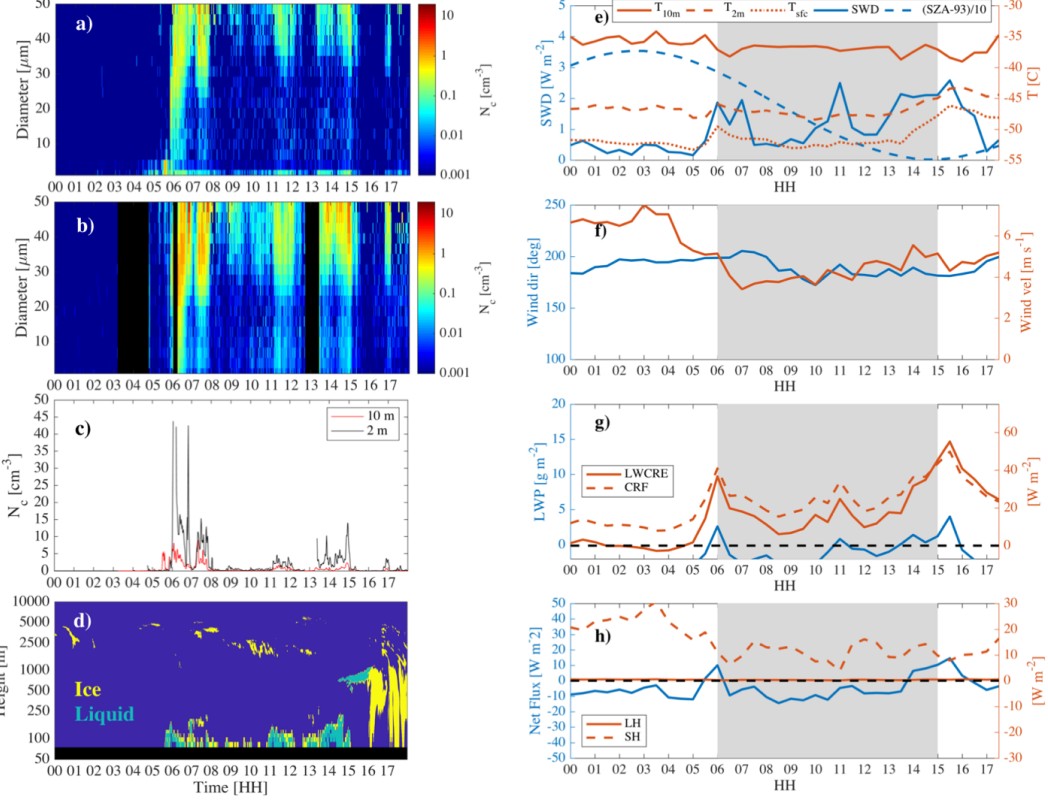

**Figure 8.** 16 January 2014 case study. Number concentration from FM100 at 10 m (a) and 2 m (b). (c) total number concentration ($N_c$) in all bins for the 10 m FM100 (red) and the 2 m FM100 (black). (d) cloud mask from the MPL: yellow is ice, green is liquid, blue is clear and black is below the minimum height for acceptable data. The height scale is log. (e) Temperatures at surface (skin), 2 m and 10 m (reds) and downwelling shortwave radiation (SWD), net shortwave radiation (SW$_{net}$) and solar zenith angle (minus 93 then divided by 10 to fit on the figure) (blues). (f) wind direction (blue) and velocity (red). (g) longwave cloud radiative effect (LWCRE) and total cloud radiative forcing (CRF) (reds) and liquid water path (LWP) (blue). (h) latent (LH) and sensible (SH) (reds) turbulent fluxes and net flux (blue).



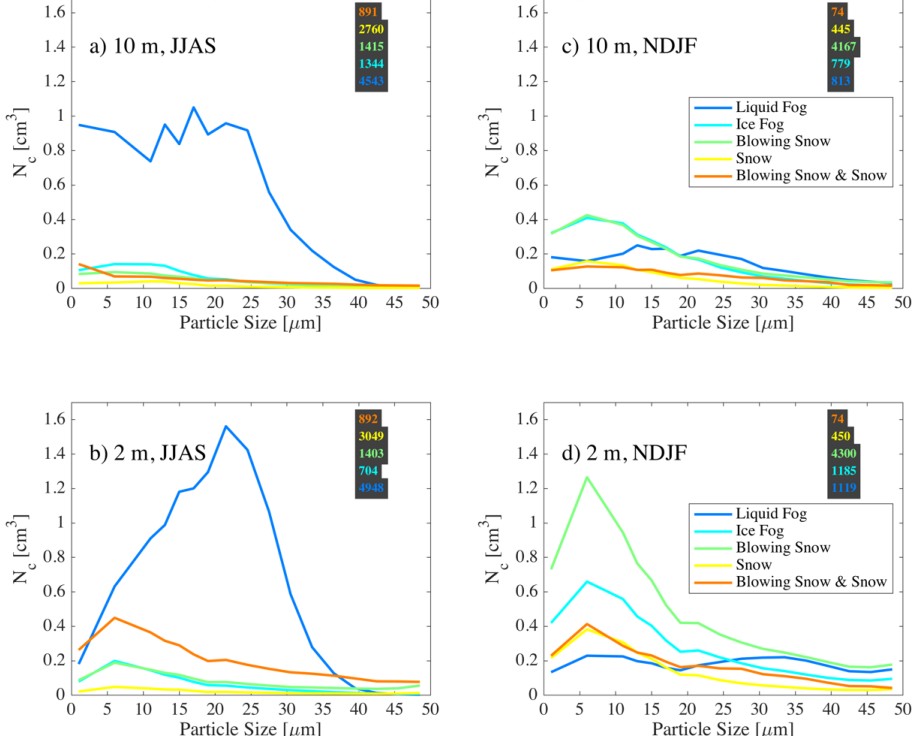

**Figure 9.** Average number concentration ($N_c$) within the FM100 size bins for different classifications measured at 10 m (top row; a, c) and 2 m (bottom row; b, d). The left column (a,b) is for June-September (JJAS) and the right column (c,d) is for November-February (NDJF). The bin centres for the sizes are 1, 6, 11, 13, 15, 17, 19, 21.5, 24.5, 27.5, 30.5, 33.5, 36.5, 39.5, 42.5, 45.5 and 48.5 μm (refer also to the Supplement for additional information on FM100 sizing). Because the FM100 bin sizes are variable, the bin counts have been normalized such that the integral of the curves = average concentration for all bins. The values over the grey background are the number of 1-minute samples in each distribution.



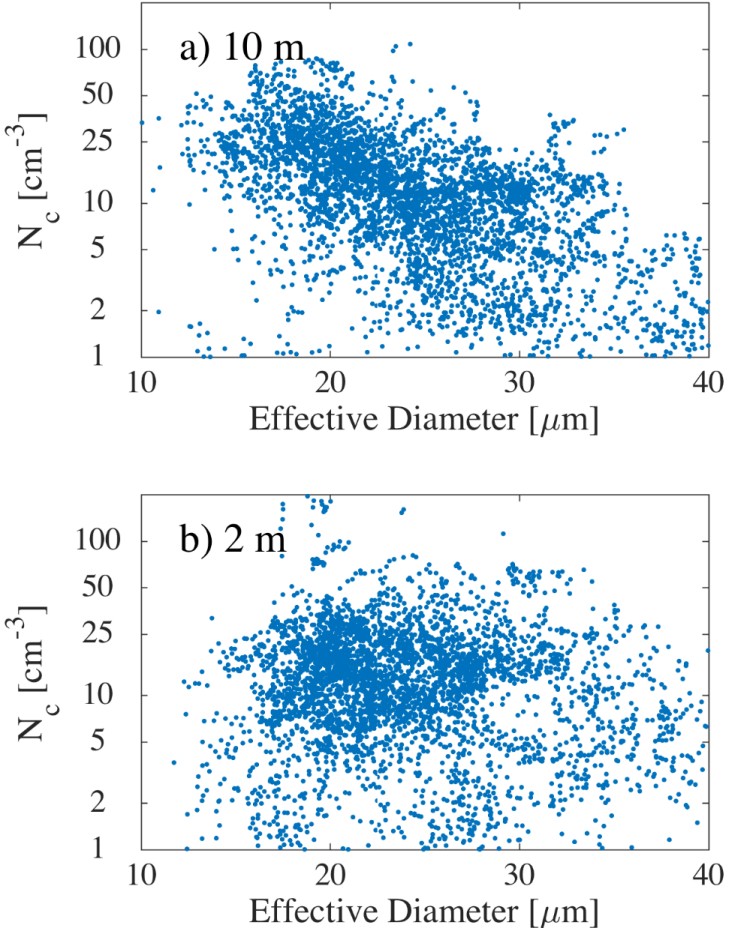

**Figure 10.** Number concentration ($N_c$) as a function of particle size for liquid fogs measured by FM100s at 10 m (a) and 2 m
(b). Note that the y-axis is log.





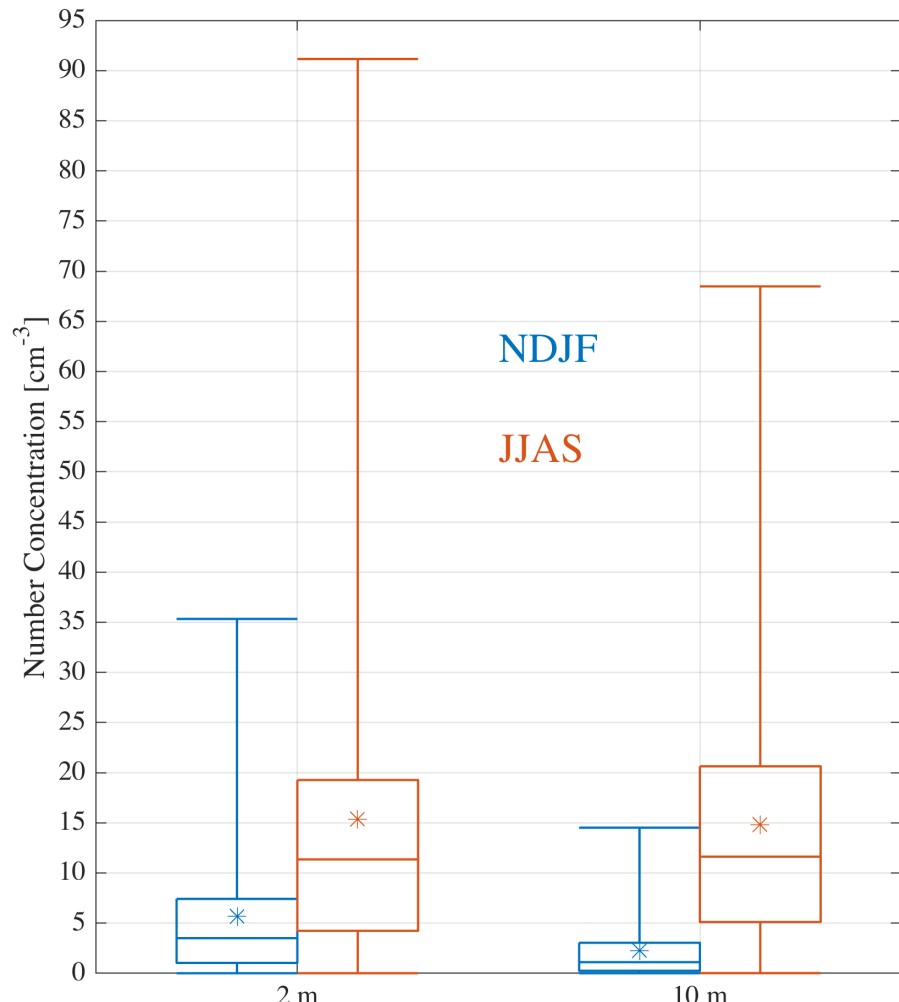

**Figure 11.** Box and whisker plots (* = mean, boxes are 25th, 50th, 75th percentile, whiskers are 1st and 99th percentiles) for all observations at times when both probes were operational in winter (NDJF, blue) and summer (JJAS, red) at 2 m and 10 m.



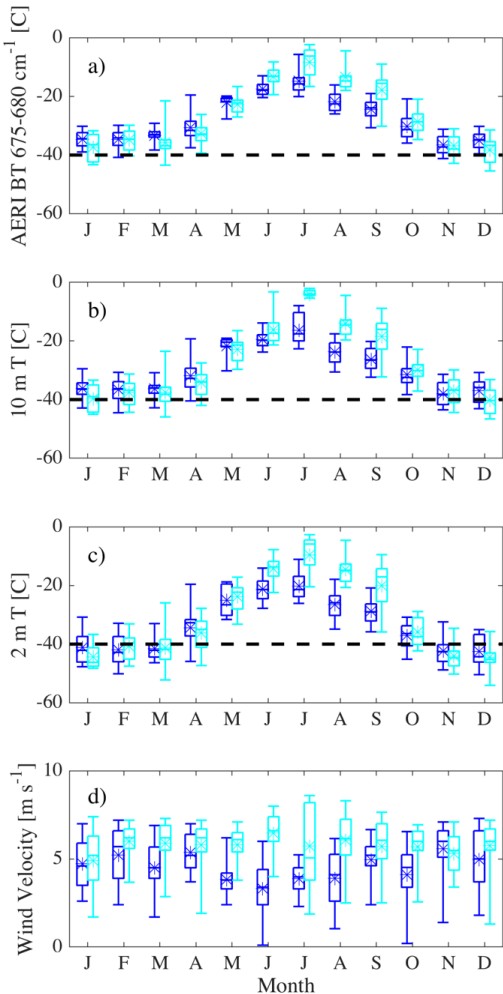

**Figure 12.** Box and whisker plots (* = mean, boxes are 25th, 50th, 75th percentile, whiskers are 5th and 95th percentiles) for each month for the ice fog category (cyan) and liquid fogs (blue). (a) AERI 675-680 cm⁻¹ brightness temperatures (BT), (b) 10 m air and (c) 2 m air. (d) is wind velocity at 10 m.




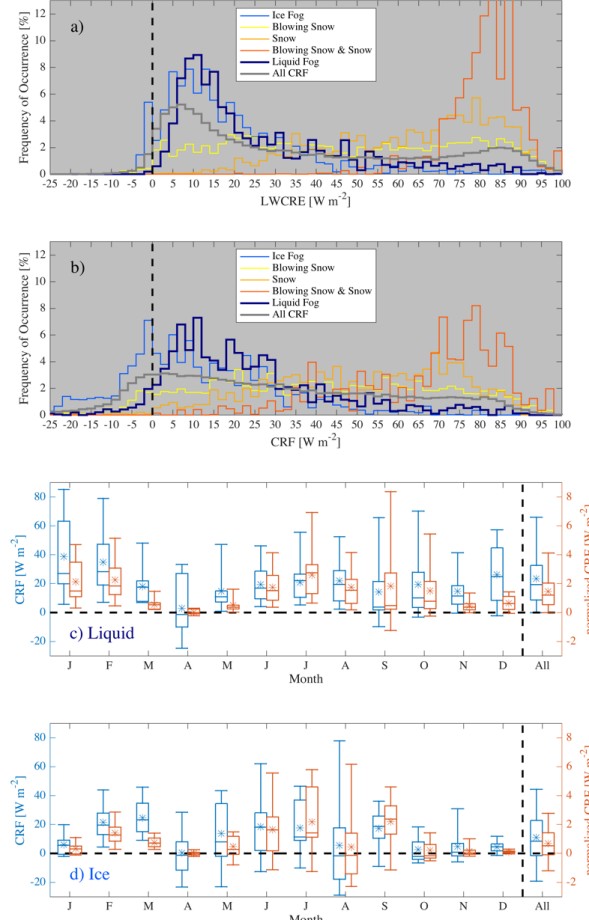

**Figure 13.** Distributions for all seasons for classified data types of (a) longwave cloud radiative effect (LWCRE), defined as

5  the perturbation to the downwelling longwave radiation (LWD) cause by clouds, LWD – LWD$_{clear-sky}$. (b) cloud radiative forcing

(CRF). (c) for just liquid fog classifications, box and whisker plots (* = mean, boxes are 25[th], 50[th], 75[th] percentile, whiskers are

5[th] and 95[th] percentiles) for each month for CRF of identified cases (blue) and CRF normalized by frequency of occurrence of

liquid fog (red). (d) as in (c), but for the ice fog category.