# Peer review of "Super-cooled liquid fogs over the central Greenland ice sheet"

_Atmospheric Chemistry and Physics, 2018_

## Referee Comment (RC1) · Anonymous Referee #1 · 13 Nov 2018

This article provides some interesting datasets about the physical nature of fogs and their radiative forcing that are worthwhile of publication. Aside from technical and scientific concerns outlined below, the primary criticism is that the writing quality is of poor quality, and this obscures the potential impact of the results. It suffers from "rambling", lacking concision that leads the reader clearly from the introduction to the conclusions. It contains many sentences that are nearly impossible to follow, statements that lack justification, and the sin of using non-quantitative descriptors like "very" and "quite". It is not clear that anyone other than the lead author read the manuscript in detail.

Specific items

1. p.2 l. 19. "liquid more efficiently absorbs and emits longwave radiation than ice likely increasing the forcing from liquid fog per unit mass compared to ice fogs or clear-sky

ice precipitation". Avoid words like "likely". Otherwise, what does this mean without stating the wavelength band? Is this really true? Does it even matter that much given the size of particles is the dominant consideration determining the specific absorption?

2. p.4 l. 26 No mention is made of size ambiguities by forward scattering probes to the assumed shape of particles. The relationship between integrated intensity in angular regime considered as a function of size is a strong function of ice crystal shape model and can result in uncertainties in measured size much greater than 100%.

3. I generally cannot follow the writing or justifications for the methods outlined in Chapter 3. This section in particular needs careful editing.

4. p. 6 l. 25 How is it known that the clouds are single layer?

5. p. 7 l. 16 I do not see any reference in Shupe et al. (2013) to justify the systematic statement that reflectivities >-5 dBz are indicative of snow.

6. What justification is there to support that blowing snow does not extend above 300 m?

7. What is the "lofting parameterization"?

8. When observations are mentioned, mention the instrument used. Presumably it is the FM100 that was used to observe particles in the lowest 10 m? Say so.

9. "Recall from Figure 3 that the threshold for identification used to construct Figure 5 is small (10-3 cm-3). Figure 3 also shows that the threshold at which events begin to be missed, and the rate at which missed events increase, is different for each class. For example, as expected, low density types such as snow also require a low threshold to be captured while high density types such as the fogs are relatively insensitive to the threshold." Completely incomprehensible.

10. p. 7 l. 4 "Snow occurred more often without significant blowing snow" ??? Writing

11. Case studies. An allowance needs to be made for observed changes in atmospheric and fog state being due to advection into the observation region rather than the apparent default assumption that the changes are evolutions within the region due to local physical processes.

12. p. 10 "Thus, while the fog was likely induced by radiation initially, it was maintained, and ultimately continued to grow without additional infrared loss at the surface driving saturation in the air column" The sentences that follow are generally incomprehensible but I do not understand how it could be that cloud top continued radiative cooling would not be able to maintain and deepen the fog layer. If the boundary layer cools, then the saturation mixing ratio decreases and the condensate mixing ratio increases.

13. p. 12 l. 25. Referencing Hansen and Travis (1974), while correct, is a bit unnecessary given how far removed that paper is from this one. Just mention the effective diameter as being the ratio of moments of the size distribution.

14. The fog bows are interesting. The physics should be described, noting in particular that the presence of fog bows does not exclude the presence of ice, only requiring the presence of liquid.

---

## Referee Comment (RC2) · Anonymous Referee #2 · 21 Dec 2018

The paper provides a detailed analysis of fog hydrometeors and their effects at the Summit site from 2012 to 2014. Interesting results include the identification of liquid versus ice fogs, their microphysical properties including their time evolution and seasonality, how ice particles and super-cooled droplets can co-exist in a vertical column in the boundary layer, and their effects on radiative forcing. The paper is well-written and organized, the measurements are of scientific significance and the novel results improve our understanding of the role of cold fogs in an Arctic setting. I would accept the paper as it stands but ask the authors to address the following minor questions.

1. A fairly convincing case is made for the settling of hydrometers to explain the difference of microphysical properties between 10 and 2 m in some cases. Based on the mean particle size can you calculate the terminal velocity and see if it is consistent with the time lag identified at 2 m with respect to 10 m?

2. Although the Summit site is likely quite flat, can the authors rule out the possibility of any uplift effects when the wind direction and speed might favor adiabatic cooling based on the local but subtle topography.

3. Is it possible that thermal tides, as measured by the surface barometer, might have a role in the observed diurnal signals when solar radiation is not the obvious forcing mechanism?

---

## Author Comment (AC1) · 31 Jan 2019

Thank you to the Reviewers for providing constructive comments, which have helped us to improve the manuscript. Our point-by-point responses are provided below in blue text following each of the Reviewer comments, shown in black.

Reviewer #1

This article provides some interesting datasets about the physical nature of fogs and their radiative forcing that are worthwhile of publication. Aside from technical and scientific concerns outlined below, the primary criticism is that the writing quality is of poor quality, and this obscures the potential impact of the results. It suffers from "rambling", lacking concision that leads the reader clearly from the introduction to the conclusions. It contains many sentences that are nearly impossible to follow, statements that lack justification, and the sin of using non-quantitative descriptors like "very" and "quite". It is not clear that anyone other than the lead author read the manuscript in detail.

We appreciate that the reviewer recognizes the value of this study. We have addressed the technical concerns as well as the specific stylistic and editorial comments itemized below. We have also edited for grammar, reduced unnecessary words and asides within the text, and removed qualitative descriptors such as those identified, as well as others (e.g., "fairly" and "relatively"). We edited the text throughout, but focused on Section 3 in accordance with the recommendations in Comment 3.

Specific items

1. p.2 l. 19. "liquid more efficiently absorbs and emits longwave radiation than ice likely increasing the forcing from liquid fog per unit mass compared to ice fogs or clear-sky ice precipitation". Avoid words like "likely". Otherwise, what does this mean without stating the wavelength band? Is this really true? Does it even matter that much given the size of particles is the dominant consideration determining the specific absorption?

Thank you for highlighting this sentence as problematic. Phase does influence absorptivity, but it is dependent on wavelength, as you say (see e.g., Fig. 1 in Turner 2003 [doi:10.1175/1520-0450(2003)042<0701:CPDUGA>2.0.CO;2]). The word "likely" qualifies the second clause of the sentence as a hypothesis. However, we do not test this hypothesis directly in this study. The intended purpose of the statement was to alert the reader to the fact that the microphysical and radiative properties of clouds are linked. We have reworked the sentence to be more generalized and to remove the speculative clause. We have also reordered the structure of the paragraph to improve the flow.

2. p.4 l. 26 No mention is made of size ambiguities by forward scattering probes to the assumed shape of particles. The relationship between integrated intensity in angular regime considered as a function of size is a strong function of ice crystal shape model and can result in uncertainties in measured size much greater than 100%.

The main focus of this study is on fogs that are composed of spherical liquid droplets. The sizes calculated from the FM100 measurement should be interpreted as optically-equivalent spheres. For non-spherical ice, the sizes calculated from the measurements do indeed have large uncertainties. Therefore, data collected when fogs were composed of ice is shown only in Figure 9 and only for contrast to the liquid cases. In the section of the paper where Figure 9 is discussed,

we actually did note the errors associated with the sizing of ice. We felt that these errors were sufficiently important for the reader to understand that we noted them when Figure 9 was introduced instead of earlier within the methods section. Specifically, at the beginning of Section 5.1, we stated the following: *Ice particles are non-spherical and, thus, the distributions represent an effective size with reference to the scattering properties of spherical liquid; asphericity and orientation are important factors in the sizing of ice using a scattering spectrometer that impose significant uncertainties (Borrmann et al., 2000). Thus, the distributions of ice classes should be treated cautiously and are shown here for context.*

Section 3 (methods) already indicates how the calculation should be interpreted, but we agree that an explicit acknowledgement of non-spherical size uncertainties is warranted there too. Therefore, we have kept the original statements in Section 5.1, but have also expanded the discussion in Section 3 and included the Borrmann et al. reference there as well.

3. I generally cannot follow the writing or justifications for the methods outlined in Chapter 3. This section in particular needs careful editing.

We have edited this section. We have also restructured the organization to be consistent with the classification procedure and positioned these steps under subheadings.

4. p. 6 l. 25 How is it known that the clouds are single layer?

This is a reasonable assumption because we focus on fogs occurring during otherwise clear skies, but it is an assumption nonetheless. We have clarified the statement accordingly.

5. p. 7 l. 16 I do not see any reference in Shupe et al. (2013) to justify the systematic statement that reflectivities >-5 dBz are indicative of snow.

This is stated in the caption for Figure 14 in Shupe et al. (2013). This is a conservative estimate relative to Shupe et al. (2007; doi 10.1029/2007GL031008), a threshold that Shupe et al. (2013) refers to as "light precipitation" (note that it does not rain at Summit). We have clarified this in the text.

6. What justification is there to support that blowing snow does not extend above 300 m?

We do not have detailed estimates of the depth of blowing snow plumes in Greenland. However, some work has been done in Antarctica, which is the closest analogue available. Distributions of blowing snow from two Antarctic stations presented in Figure 11 of Gossart et al. (2017, doi: 10.5149/tc-11-2755-2017) show nearly all events to be less than 300 m depth. Gossart et al. reported that some events were much deeper (heights above 1000 m), but these were the exception rather than the norm and were generally found to coincide with precipitation. We have added the Gossart et al. reference to 3.2.3.

7. What is the "lofting parameterization"?

The reference is Li and Pomeroy (1997), as described in the sentence prior to the phrase in question. To improve clarity, we have replaced "lofting parameterization" with "Li and Pomeroy parameterization" and clarified the terms "lofting" and "blowing".

8. When observations are mentioned, mention the instrument used. Presumably it is the FM100 that was used to observe particles in the lowest 10 m? Say so.

We believe this comment references the distinction between the 2 m FM100 and the 10 m FM100 rather than the distinction between the FM100s and other instruments, but it is unclear from the comment precisely where in the text this ambiguity was identified. Generally, the distinctions are clearly stated in the text where necessary, as well as in all relevant figures/captions. In some parts of the manuscript, such as the section that describes the processing of the FM100 data, no such distinction is made, but we feel that it is sufficiently clear that we are referring to both probes. We have clarified one statement where we felt there was ambiguity in the list that appears at the end of Section 3.

9. "Recall from Figure 3 that the threshold for identification used to construct Figure 5 is small (10-3 cm-3). Figure 3 also shows that the threshold at which events begin to be missed, and the rate at which missed events increase, is different for each class. For example, as expected, low density types such as snow also require a low threshold to be captured while high density types such as the fogs are relatively insensitive to the threshold." Completely incomprehensible.

This sentence was intended to demonstrate support for the classification methodology by explaining that the behavior of the lines in Figure 5 is consistent with expectations for low density (e.g., snow) and high density (e.g., fog) events. This is not a critical statement and it was removed when we edited Section 3 in response to Comment 3 and your Summary comments.

10. p. 7 l. 4 "Snow occurred more often without significant blowing snow" ??? Writing

We have rewritten the sentence in question.

11. Case studies. An allowance needs to be made for observed changes in atmospheric and fog state being due to advection into the observation region rather than the apparent default assumption that the changes are evolutions within the region due to local physical processes.

We have added this qualification to the introductory statements at the top of Section 4. Reviewer #2 asked a similar question about the effects of local topography, which we have also acknowledged in the same place.

12. p. 10 "Thus, while the fog was likely induced by radiation initially, it was maintained, and ultimately continued to grow without additional infrared loss at the surface driving saturation in the air column" The sentences that follow are generally incomprehensible but I do not understand how it could be that cloud top continued radiative cooling would not be able to maintain and deepen the fog layer. If the boundary layer cools, then the saturation mixing ratio decreases and the condensate mixing ratio increases.

The sentence in question refers to cooling from the surface, not the cloud top. We know that the surface stopped cooling because we observed the near-surface air to warm due to increased radiative forcing from the fog. Cloud top cooling is suggested later as a plausible mechanism to introduce moisture to the surface in a manner consistent with your statements. We have edited the sentences following the one in question. We acknowledge that one of these sentences (beginning "While…") was fragmented and we have fixed this. We have also added clarity and we hope that this addresses your concerns.

13. p. 12 l. 25. Referencing Hansen and Travis (1974), while correct, is a bit unnecessary given how far removed that paper is from this one. Just mention the effective diameter as being the ratio of moments of the size distribution.

We have made this change.

14. The fog bows are interesting. The physics should be described, noting in particular that the presence of fog bows does not exclude the presence of ice, only requiring the presence of liquid.

We have added a brief explanation and a reference for the optical physics of fogbows. You are correct that the occurrence of the fogbow does not rule out ice. We appreciate this suggestion and have added the qualification. We have also added an additional qualification (with references) indicating that quasi-spherical ice has been observed by others in other locations. This fact implies that we should also not rule out the possibility that under certain conditions ice fogs might produce optical phenomena similar to what is regularly observed in liquid fogs. However, this possibility is speculative for the following reasons: First, the science on habits for ice fogs is not settled (e.g., Gultepe et al. 2015, doi: 10.1016/j.atmosres.2014.04.014); second, regions such as Fairbanks, Alaska, where much of the literature on ice fogs originates is a poor analogue for Greenland due to differences in ice nucleating aerosols; and third, we are not aware of any literature specifically linking optical phenomena normally associated with liquid droplets to ice fogs.

Reviewer #2

The paper provides a detailed analysis of fog hydrometeors and their effects at the Summit site from 2012 to 2014. Interesting results include the identification of liquid versus ice fogs, their microphysical properties including their time evolution and seasonality, how ice particles and super-cooled droplets can co-exist in a vertical column in the boundary layer, and their effects on radiative forcing. The paper is well-written and organized, the measurements are of scientific significance and the novel results improve our understanding of the role of cold fogs in an Arctic setting. I would accept the paper as it stands but ask the authors to address the following minor questions.

We appreciate the reviewer's positive feedback. We found the questions below to be intriguing and have endeavored to address them.

1. A fairly convincing case is made for the settling of hydrometers to explain the difference of microphysical properties between 10 and 2 m in some cases. Based on the mean particle size can you calculate the terminal velocity and see if it is consistent with the time lag identified at 2 m with respect to 10 m?

Following your suggestion, we calculated a time series of theoretical particle terminal velocities for the summertime case study, which is the case we used to estimate the settling rates you reference from the manuscript. We made these calculations following Pruppacher and Klett (2010, doi: 10.1007/978-0-306-48100-0) and the results are displayed in the figure below. Two calculations are shown: The first is based on the simple assumption of the Stokes regime (blue in the figure) and the second is corrected for slip-flow effects, which become important for small particles (red in the figure). All particles are assumed to be spherical liquid droplets and the flow is assumed to be laminar. The two theoretical calculations are similar and agree with the velocities we derived from the probes. Specifically, the theoretical calculations indicate settling rates between 0 and 0.01 m/s during the first hour of the case when we estimated ~0.01 m/s using the probe data and 0.03-0.035 m/s during the mature phase of the fog when the probes suggested a settling rate of 0.03-0.04 m/s. We decided to show the new figure here to document our work to

address this question, but have elected not to add it to the manuscript. Instead, we have added two sentences to the text explaining that the settling rates implied by the time series derived from the probes are consistent with theoretical calculations, referring readers to Pruppacher and Klett (2010).

[Figure]

2. Although the Summit site is likely quite flat, can the authors rule out the possibility of any uplift effects when the wind direction and speed might favor adiabatic cooling based on the local but subtle topography.

This is an excellent question. Reviewer #1 asked a similar question, but referring to advection from more distant regions. The area around Summit Station is indeed quite flat. A number of detailed topographic surveys have been conducted in the area by the Cold Regions Research and Engineering Laboratory (CRREL) for logistical management of the station structures. For example, see https://www.geosummit.org/sites/default/files/docs/Summit-Station_ERDC-CRREL_TR-16-16.pdf. These surveys indicate that the maximum slope is approximately 1% at the kilometer scale. It is <1% towards the south, into the predominant wind direction from which our sampling was conducted. Locally around the station, variability in surface height is ~±2-3 m and is generally more complex than the surrounding environment because of localized drifting associated with station structures. Much of our analysis of the case studies assumes relative spatial homogeneity. Unfortunately, we cannot rule out the possibility that local topographic effects (or advection from more distant areas) might explain some of the observed time-evolution of the fogs at the location of the measurements. Therefore, we have included a statement acknowledging these limitations in the introductory remarks at the beginning of the case study section.

3. Is it possible that thermal tides, as measured by the surface barometer, might have a role in the observed diurnal signals when solar radiation is not the obvious forcing mechanism?

This is a very intriguing hypothesis! We have conducted some work linking buoyancy waves above the boundary layer to fluctuations in the thermal structure of the surface layer and associated modification of the fog microphysics. This was presented at the POLAR2018 conference in June 2018 in Davos, Switzerland, and is the subject of ongoing study (AC3-2010: http://www.professionalabstracts.com/POLAR2018/iPlanner/#/grid/1529539200). Only a small portion of that work was incorporated into the current manuscript and can be found in the discussion of Figure 7. Because we have observed buoyancy waves to create areas of convergence and divergence at the surface (see Section 4.2) that also appear to modulate fog microphysics (discussed in the poster, but not in the manuscript), it is plausible that gravitational waves, such as thermal tides could be responsible and that such phenomena could trigger condensation. We revisited the POLAR2018 study and some additional cases and believe that while the occurrence of the buoyancy waves in the case study coincides with the diurnal cycle, this is incidental and not causal. We will certainly keep this idea in mind as a plausible candidate for the missing mechanism we highlight at the end of Section 4.2, but as yet we have not found supporting evidence.

---

## Author Comment (AC2) · 31 Jan 2019

**Super-cooled liquid fogs over the central Greenland ice sheet**

Christopher J. Cox12, David C. Noone4, Max Berkelhammer4, Matthew D. Shupe12, William D. Neff12, Nathaniel B. Miller4, Von P. Walden4, Konrad Steffen4

Cooperative Institute for Research in Environmental Sciences, Boulder, Colorado, 80309, USA

NOAA Earth System Research Laboratory, Boulder, Colorado, 80305, USA

College of Earth, Ocean, and Atmospheric Sciences, Oregon State University, Corvallis, Oregon, 97331, USA

Department of Earth and Environmental Sciences, University of Illinois at Chicago, Chicago, Illinois, 60607, USA

· Department of Civil and Environmental Engineering, Washington State University, Pullman, Washington, 99164, USA

4 Swiss Federal Research Institute WSL, Birmensdorf, CH-8903, Switzerland

10

5

Correspondence to: Christopher J. Cox (christopher.j.cox@noaa.gov)

**Abstract.** Radiation fogs at Summit, Greenland (72.58°N, 38.48°W, 3210 masl) are frequently reported by observers. The fogs are often accompanied by fogbows, indicating the particles are composed of liquid and because of the low temperatures at Summit, this liquid is super-cooled. Here we analyse the formation of these fogs as well as their physical and radiative

- 15 properties. In situ observations of particle size and droplet number concentration were made using scattering spectrometers near 2 m and 10 m height from 2012 to 2014. These data are complemented by co-located observations of meteorology, turbulent and radiative fluxes, and remote sensing. We find that liquid fogs occur in all seasons with the highest frequency in September and a minimum in April. Due to the characteristics of the boundary-layer meteorology, the fogs are elevated, forming between 2 m and 10 m and the particles then fall toward the surface. The diameter of mature particles is typically 20-
- 20 25 µm in summer. Number concentrations are higher at warmer temperatures and, thus, higher in summer compared to winter. The fogs form at temperatures as warm as ~5 °C, while the coldest form at temperatures approaching -40 °C. Facilitated by the elevated condensation, in winter 2/3 of fogs occurred within a relatively warm layer above the surface when the near-surface air was below -40 °C, as cold as -57 °C, which is too cold to support liquid water. This implies that fog particles settling through this layer of cold air freeze in the air column before contacting the surface, thereby accumulating at the surface as ice
- 25 without riming. Liquid fogs observed under otherwise clear skies imparted annually 1.5 W m2 of cloud radiative forcing (CRF). While this is a small contribution to the surface radiation climatology, individual events are influential. The mean CRF during liquid fog events was 26 W m2, and was sometimes be much higher. An extreme case study was observed to radiatively force 5 °C of surface warming during the coldest part of the day, effectively damping the diurnal cycle. At lower elevations of the ice sheet where melting is more common, such damping could signal a role for fogs in preconditioning the surface for melting later in the day.

[revised manuscript text omitted]

|---|-----------------------------------------------------------------------------------------------------------------------------------------------------------------------------------------------------------------------------------|
| ή | Deleted: consists of five steps                                                                                                                                                                                                   |
| 4 | Deleted: elevated                                                                                                                                                                                                                 |
| 4 | Deleted: with bases above the surface                                                                                                                                                                                             |
|   | Deleted: Finally, (5), (1)-(4) are combined optimally.                                                                                                                                                                     |
|   | Formatted: Font: Bold                                                                                                                                                                                                             |
| - | Formatted: Font: Bold                                                                                                                                                                                                             |
|   | Deleted: which include observations that are not traditionally classified as clouds,                                                                                                                                       |
| 4 | Deleted: ,                                                                                                                                                                                                                        |
|   | Deleted: , and transient particles, but also include forming and dissipating fogs                                                                                                                                          |
| Ì | Deleted: various                                                                                                                                                                                                                  |
|   | Deleted: P                                                                                                                                                                                                                        |
| 4 | Deleted: of any type are measured                                                                                                                                                                                          |
| Ì | Deleted: observed                                                                                                                                                                                                                 |
| Ò | Deleted: between                                                                                                                                                                                                                  |
| Q | Deleted: and 90%                                                                                                                                                                                                                  |
| Ņ | Deleted: , but there is relatively little sensitivity to                                                                                                                                                                          |
|   | Deleted: s                                                                                                                                                                                                                        |
| N | Deleted: of order                                                                                                                                                                                                                 |
| Ŋ | Deleted: or smaller                                                                                                                                                                                                               |
| l | Deleted: lower                                                                                                                                                                                                                    |
| Ì | Formatted: Font: Bold                                                                                                                                                                                                             |
| h | Formatted: Font: Bold                                                                                                                                                                                                             |
| γ | Formatted: Indent: First line: 0.5"                                                                                                                                                                                               |
|   |                                                                                                                                                                                                                                   |

[revised manuscript text omitted]

---

## Author Response (AR2)

Thank you to the Reviewers for providing constructive comments, which have helped us to improve the manuscript. Our point-by-point responses are provided below in blue text following each of the Reviewer comments, shown in black. The line numbers referenced at the end of each comment refer to the line numbers in acp-2018-819-AC2-supplement.pdf, the pdf-version of the tracked changes document that is available as AC2, uploaded to the public discussion by the Author on January, 31, 2019 at the following url: https://www.atmos-chem-phys-discuss.net/acp-2018-819/. Note that these line numbers will differ from the merged (response/manuscript) document that was requested separately by ACP and may have been provided to the referees.

Reviewer #1

This article provides some interesting datasets about the physical nature of fogs and their radiative forcing that are worthwhile of publication. Aside from technical and scientific concerns outlined below, the primary criticism is that the writing quality is of poor quality, and this obscures the potential impact of the results. It suffers from "rambling", lacking concision that leads the reader clearly from the introduction to the conclusions. It contains many sentences that are nearly impossible to follow, statements that lack justification, and the sin of using non-quantitative descriptors like "very" and "quite". It is not clear that anyone other than the lead author read the manuscript in detail.

We appreciate that the reviewer recognizes the value of this study. We have addressed the technical concerns as well as the specific stylistic and editorial comments itemized below. We have also edited for grammar, reduced unnecessary words and asides within the text, and removed qualitative descriptors such as those identified, as well as others (e.g., "fairly" and "relatively"). We edited the text throughout, but focused on Section 3 in accordance with the recommendations in Comment 3 (for line numbers, please refer to our response to Comment 3, below).

Qualitative descriptors were removed in the following locations:
Page 1, Line 26: removed "relatively"
Page 5, Line 4: changed "which is fairly stable" to "which varies"
Page 7, Line 19: removed "quite"
Page 7, Line 20: removed "very"
Page 7, Line 21: removed "very"
Page 7, Line 23: removed "generally"
Page 7, Line 24: removed "quite"
Page 7, Line 25: removed "relatively"
Page 8, Line 24: removed "indeed"
Page 10, Line 7: removed "relatively"
Page 10, Line 26: removed "strong"
Page 11, Line 6: removed "indeed"
Page 11, Line 19: removed "very"
Page 11, Line 25: removed "considerably"
Page 13, Line 11: removed "quite"
Page 14, Line 11: removed "fairly evenly"
Page 14, Line 27: removed two instances of "very"
Page 15, Line 15: removed "relatively"
Page 16, Lines 5: removed "relatively"

Page 16, Line 8: removed "intricate"

Edits for tense:
Page 1, Line 23: changed "is" to "was"
Page 1, Line 25: added "observed" and changed "impart" to "imparted"
Page 1, Line 27: changed "is" to "was" and "but can" to "and was"
Page 9, Line 28: changed "observable" to "observed"
Page 11, Line 15: changed "fog that is" to "particles"
Page 11, Line 16: changed "is the fog that was" to "were"
Page 12, Line 31: added "show" and "that" to make the sentence active present

Grammar:
Page 1, Line 23: changed "which is well below that which can" to "too cold to"
Page 4, Line 14: removed comma
Page 4, Line 15: changed "in each probe's inlet tunnel, was approximately" to "located in the probe's inlet tunnel; at Summit, the PAS was approximately"
Page 5, Line 24: removed comma and moved "as described by Miller et al. (2015) from the middle of the sentence to the end of the sentence
Page 9, Line 12: added missing word "further"
Page 9, Line 15: removed extra space
Page 12, Line 22: changed "thus" to "specifically"
Page 12, Line 31: changed colloquial "When considering the" to "The", removed a comma
Page 18, Line 7: changed font to comply with ACP style
Page 19, Line 17: removed a comma to comply with ACP style
Page 19, Line 26: changed font to comply with ACP style

Specific items

1. p.2 l. 19. "liquid more efficiently absorbs and emits longwave radiation than ice likely increasing the forcing from liquid fog per unit mass compared to ice fogs or clear-sky ice precipitation". Avoid words like "likely". Otherwise, what does this mean without stating the wavelength band? Is this really true? Does it even matter that much given the size of particles is the dominant consideration determining the specific absorption?

Thank you for highlighting this sentence as problematic. Phase does influence absorptivity, but it is dependent on wavelength, as you say (see e.g., Fig. 1 in Turner 2003 [doi:10.1175/1520-0450(2003)042<0701:CPDUGA>2.0.CO;2]). The word "likely" qualifies the second clause of the sentence as a hypothesis. However, we do not test this hypothesis directly in this study. The intended purpose of the statement was to alert the reader to the fact that the microphysical and radiative properties of clouds are linked. We have reworked the sentence to be more generalized and to remove the speculative clause. We have also reordered the structure of the paragraph to improve the flow.

The new line numbers for these changes are Page 2, Lines 18-24. Specifically,

- The statement in question now reads as follows: "Because cloud microphysical and radiative properties are linked (e.g., Garret and Zhao, 2006; Shupe and Intrieri, 2004), microphysical observations of the fogs at Summit are needed to better

understand their radiative forcing and to provide constraints for modeling." See Page 2, Lines 19-21.

- One sentence was moved to Page 5, Lines 31-32 where it became parenthetic and an unnecessary reference was removed. The statement now reads as follows: "(note that we do not distinguish between ice fog and clear-sky precipitation, also known as "diamond dust")".
- The reference to Intrieri and Shupe (2004), removed from the statement on diamond dust, was removed from the reference list on Page 19, Line 27
- The first sentence of the paragraph was moved to the end of the paragraph to improve flow
- "Despite being optically thin" was removed at Page 2, Line 21

2. p.4 l. 26 No mention is made of size ambiguities by forward scattering probes to the assumed shape of particles. The relationship between integrated intensity in angular regime considered as a function of size is a strong function of ice crystal shape model and can result in uncertainties in measured size much greater than 100%.

The main focus of this study is on fogs that are composed of spherical liquid droplets. The sizes calculated from the FM100 measurement should be interpreted as optically-equivalent spheres. For non-spherical ice, the sizes calculated from the measurements do indeed have large uncertainties. Therefore, data collected when fogs were composed of ice is shown only in Figure 9 and only for contrast to the liquid cases. In the section of the paper where Figure 9 is discussed, we actually did note the errors associated with the sizing of ice. We felt that these errors were sufficiently important for the reader to understand that we noted them when Figure 9 was introduced instead of earlier within the methods section. Specifically, at the beginning of Section 5.1 (Lines 5-8 of the tracked-changes document) we stated the following: *Ice particles are non-spherical and, thus, the distributions represent an effective size with reference to the scattering properties of spherical liquid; asphericity and orientation are important factors in the sizing of ice using a scattering spectrometer that impose significant uncertainties (Borrmann et al., 2000). Thus, the distributions of ice classes should be treated cautiously and are shown here for context.*

Section 2 (Page 4, Lines 18-19) already indicates how the calculation should be interpreted, but we agree that an explicit acknowledgement of non-spherical size uncertainties is warranted there too. Therefore, we have kept the original statements in Section 5.1, but have also expanded the discussion in Section 2 and included the Borrmann et al. reference there as well. This change can be found on Page 4, Lines 23-24. Specifically, we changed "Thus, for the present work, the term particle size refers to the measurement of individual hydrometeors averaged over measurements made 60 times each minute and should be interpreted as an optically-equivalent diameter" to ""Thus, for the present work, the term particle size refers to the measurement of individual hydrometeors averaged over measurements made 60 times each minute and should be interpreted as an optically-equivalent diameter of spheres, regardless of the particles geometric shape. Refer to Borrmann et al. (2000) for information on sizing errors associated with ice spheres."

3. I generally cannot follow the writing or justifications for the methods outlined in Chapter 3. This section in particular needs careful editing.

We have edited this section. We have also restructured the organization to be consistent with the classification procedure and positioned these steps under subheadings. The line numbers for these changes begin on Page 5, Line 30 and extend to Page 8, Line 14. Specifically:

- The new headings are described below with the list of changes to the text organized within each.
  - 3.1. Introduction to the classification (Page 5, Line 30)
    - Page 5, Lines 31-32: The qualification about diamond dust was moved here from the introduction because it is more important to this section than to the introduction. See also our response to your Comment #1.
    - Page 6, Line 1: "subset" changed "identify"
    - Page 6, Line 2: Removed unnecessary statement beginning "This information comes from…"
    - Page 6, Line 3: changed "consists of five steps" to "is as follows"
    - Page 6, Lines 4-5: clarified step 2 as "elevated tropospheric cloud layers" in place of "bases above the surface"
    - What was formerly Step 5 ("Steps 1-4 are combined optimally") at Page 6, Line 6, was rewritten so that the manner in which steps 1-3 are combined with 4 is explicitly stated, referencing each of the steps. The new phrasing appears on Page 6, Lines 5-7: "(4) the phase of the particles is determined for all cases where particles were observed near the surface (step 1) but the sky was otherwise clear (step 2) and there was not blowing snow (step 3)."
  - 3.2.1. Identification of near-surface particles (Page 6, Line 8)
    - Unnecessary and vague phrasing was removed on Page 6, Lines 9-10
    - The final sentence of the subsection was edited for clarity (Page 6, Lines 14-15)
  - 3.2.2. Tropospheric clouds (Page 6, Line 17)
    - This section was not edited, but was moved to appear earlier in the section to be consistent with the ordering of the steps listed in 3.1.
  - 3.2.3. Snow and blowing snow (Page 6, Line 22)
    - As with 3.2.2, the text in 3.2.3 was moved to appear earlier in the section to be consistent with the ordering of the steps listed in 3.1. The only changes to this section are those made in response to Comment #6 (see below).
  - 3.2.4. Phase classification (Page 6, Line 31)
    - The phase classification now appears last in Section 3.2 such that the subsection 3.2.N refer to the steps (N = 1 through 4) listed in 3.1.
    - Edits were made to lines 9 and 10 Page 7 in response to Comment #4, below.
    - Edits were made to grammar and to remove qualitative descriptors in response to the main comment at the beginning of the review.
    - The deletion of the text that was moved to becomes sections 3.2.2 and 3.2.3 can be found on Page 7, Line 27.
  - 3.3. Summary of classification (Page 7, Line 28)

- A sentence was removed at Page 8, Line 3 that caused confusion and was the subject of Comment #9. Refer also to our response to Comment #9.
- A statement was moved from Page 8, Line 15 to appear earlier in 3.3 beginning at Page 7, Line 33. This statement was also simplified.
- Bullet-point #2 (Page 8, Line 8) was edited in response to Comment #10. Please refer to our response to that comment.

4. p. 6 l. 25 How is it known that the clouds are single layer?

This is a reasonable assumption because we focus on fogs occurring during otherwise clear skies, but it is an assumption nonetheless. We have clarified the statement accordingly. Please see Lines 9 and 10 of Page 7.

The sentence now reads as follows: "Finally, because the scenes that are tested feature fogs that were observed when the sky was otherwise clear, it is reasonable to assume that these scenes were single-layer, negating ambiguity from multiple cloud layers."

5. p. 7 l. 16 I do not see any reference in Shupe et al. (2013) to justify the systematic statement that reflectivities >-5 dBz are indicative of snow.

This is stated in the caption for Figure 14 in Shupe et al. (2013). This is a conservative estimate relative to Shupe et al. (2007; doi 10.1029/2007GL031008), a threshold that Shupe et al. (2013) refers to as "light precipitation" (note that it does not rain at Summit). We have not edited the manuscript in response to this comment. Presently, the statement can be found on Lines 23 and 24 of Page 6 where it was moved as part of the edits to Section 3 discussed above.

6. What justification is there to support that blowing snow does not extend above 300 m?

We do not have detailed estimates of the depth of blowing snow plumes in Greenland. However, some work has been done in Antarctica, which is the closest analogue available. Distributions of blowing snow from two Antarctic stations presented in Figure 11 of Gossart et al. (2017, doi: 10.5149/tc-11-2755-2017) show nearly all events to be less than 300 m depth. Gossart et al. reported that some events were much deeper (heights above 1000 m), but these were the exception rather than the norm and were generally found to coincide with precipitation.

- We have added the Gossart et al. reference to 3.2.3 on Page 6, Lines 25-27.
- The Gossart reference now appears in the reference list.
- The new sentence referencing Gossart et al. reads as follows: "While estimates of the depth of blowing snow layers are currently unavailable at Summit, in Antarctica layer depths exceeding 300 m are infrequent and when they do occur, they generally coincide with precipitation (Gossart et al., 2017)."

7. What is the "lofting parameterization"?

The reference is Li and Pomeroy (1997), as described in the sentence prior to the phrase in question. To improve clarity, we have replaced "lofting parameterization" with "Li and Pomeroy parameterization" and clarified the terms "lofting" and "blowing". Please see Page 6, Lines 28-30. These two sentences now read as follows: "The radar data are combined with wind parameterizations for lofting (i.e., blowing) snow calculated for western Canada by Li and Pomeroy (1997). When the radar data indicates snow and the Li and Pomeroy parameterization

indicates blowing snow, the classification of a combination of snow and blowing snow is assigned."

8. When observations are mentioned, mention the instrument used. Presumably it is the FM100 that was used to observe particles in the lowest 10 m? Say so.

We believe this comment references the distinction between the 2 m FM100 and the 10 m FM100 rather than the distinction between the FM100s and other instruments, but it is unclear from the comment precisely where in the text this ambiguity was identified. Generally, the distinctions are clearly stated in the text where necessary, as well as in all relevant figures/captions. In some parts of the manuscript, such as the section that describes the processing of the FM100 data, no such distinction is made, but we feel that it is sufficiently clear that we are referring to both probes. We have clarified one statement where we felt there was ambiguity in the list that appears at the end of Section 3 on Page 8, Line 11 by adding "10 m".

9. "Recall from Figure 3 that the threshold for identification used to construct Figure 5 is small (10-3 cm-3). Figure 3 also shows that the threshold at which events begin to be missed, and the rate at which missed events increase, is different for each class. For example, as expected, low density types such as snow also require a low threshold to be captured while high density types such as the fogs are relatively insensitive to the threshold." Completely incomprehensible.

This sentence was intended to demonstrate support for the classification methodology by explaining that the behavior of the lines in Figure 5 is consistent with expectations for low density (e.g., snow) and high density (e.g., fog) events. This is not a critical statement and it was removed when we edited Section 3 in response to Comment 3 and your Summary comments. The deletion can be seen on Page 8, Line 3.

10. p. 7 l. 4 "Snow occurred more often without significant blowing snow" ??? Writing

We have rewritten the sentence in question. See Page 8, Lines 8-10. The sentence now reads as follows: "In months when blowing snow occurred most frequently (winter and spring), precipitating snow was less common and during summer when most of the precipitating snow occurred was when blowing snow was least common."

11. Case studies. An allowance needs to be made for observed changes in atmospheric and fog state being due to advection into the observation region rather than the apparent default assumption that the changes are evolutions within the region due to local physical processes.

We have added this qualification to the introductory statements at the top of Section 4. Reviewer #2 asked a similar question about the effects of local topography, which we have also acknowledged in the same place. See Page 8, Lines 18-21. The two new sentences read as follows: "The discussion of the time evolution of the fogs assumes spatial homogeneity. While the meteorology observed during the cases generally supports this assumption, we cannot rule out advection, either from distant regions or associated with local topographical variability (~ ± 2-3 m) as a source of tome of the observed variability."

12. p. 10 "Thus, while the fog was likely induced by radiation initially, it was maintained, and ultimately continued to grow without additional infrared loss at the surface driving saturation in the air column" The sentences that follow are generally incomprehensible but I do not understand how it could be that cloud top continued radiative cooling would not be able to maintain and

deepen the fog layer. If the boundary layer cools, then the saturation mixing ratio decreases and the condensate mixing ratio increases.

The sentence in question refers to cooling from the surface, not the cloud top. We know that the surface stopped cooling because we observed the near-surface air to warm due to increased radiative forcing from the fog. Cloud top cooling is suggested later (Page 10, Line 7) as a plausible mechanism to introduce moisture to the surface in a manner consistent with your statements. We have edited the sentences following the one in question as follows:

- We acknowledge that one of these sentences (beginning "While…") was fragmented and we have fixed this. It now reads "The large LWP (Fig. 6f) implies significant cloud-top radiative cooling may have occurred; therefore, buoyancy-driven mixing is a plausible mechanism to have supplied the moisture that maintained the fog." See Page 10, Lines 6-9.
- We have added "relative humidity supporting the" to Page 10, Line 6 to improve clarity.
- We have improved flow by changing "The sodar data shows…" to "This is supported by the sodar data, which shows…". See Page 10, Line 9.

13. p. 12 l. 25. Referencing Hansen and Travis (1974), while correct, is a bit unnecessary given how far removed that paper is from this one. Just mention the effective diameter as being the ratio of moments of the size distribution.

We have made this change. The change can be found on Page 12, Line 25. The portion of the sentence that was changed now reads as follows: "… is supported by a calculation of the effective diameter (the ratio of the $3^{rd}$ and $2^{nd}$ moments of the size distributions) for all liquid fog scenes in all months…"

14. The fog bows are interesting. The physics should be described, noting in particular that the presence of fog bows does not exclude the presence of ice, only requiring the presence of liquid.

We have added a brief explanation and a reference for the optical physics of fogbows. You are correct that the occurrence of the fogbow does not rule out ice. We appreciate this suggestion and have added the qualification. We have also added an additional qualification (with references) indicating that quasi-spherical ice has been observed by others in other locations. This fact implies that we should also not rule out the possibility that under certain conditions ice fogs might produce optical phenomena similar to what is regularly observed in liquid fogs. However, this possibility is speculative for the following reasons: First, the science on habits for ice fogs is not settled (e.g., Gultepe et al. 2015, doi: 10.1016/j.atmosres.2014.04.014); second, regions such as Fairbanks, Alaska, where much of the literature on ice fogs originates is a poor analogue for Greenland due to differences in ice nucleating aerosols; and third, we are not aware of any literature specifically linking optical phenomena normally associated with liquid droplets to ice fogs.

- See Page 13, Lines 20-25.
- The references to Lynch and Schwartz, Thuman and Robinson and Gultepe et al. now appear in the reference list.
- The new text reads as follows: "Fogbows are formed by scattering processes dominated by diffraction when the size of spherical particles are within the Mie regime (e.g., Lynch and Schwartz 1991). Note that the presence of a fogbow is suggestive of the presence of liquid, but that is does not rule out the presence of ice. Additionally, spherical of quasi-spherical ice formed by freezing of supercooled

liquid has been reported by other studies (e.g., Thuman and Robinson 1954), though the preferred habits of ice fog particles remains controversial (see Gultepe et al. 2015) and we are unaware of any studies linking ice fogs to optical phenomena normally associated with liquid droplets."

Reviewer #2

The paper provides a detailed analysis of fog hydrometeors and their effects at the Summit site from 2012 to 2014. Interesting results include the identification of liquid versus ice fogs, their microphysical properties including their time evolution and seasonality, how ice particles and super-cooled droplets can co-exist in a vertical column in the boundary layer, and their effects on radiative forcing. The paper is well-written and organized, the measurements are of scientific significance and the novel results improve our understanding of the role of cold fogs in an Arctic setting. I would accept the paper as it stands but ask the authors to address the following minor questions.

We appreciate the reviewer's positive feedback. We found the questions below to be intriguing and have endeavored to address them.

1. A fairly convincing case is made for the settling of hydrometers to explain the difference of microphysical properties between 10 and 2 m in some cases. Based on the mean particle size can you calculate the terminal velocity and see if it is consistent with the time lag identified at 2 m with respect to 10 m?

Following your suggestion, we calculated a time series of theoretical particle terminal velocities for the summertime case study, which is the case we used to estimate the settling rates you reference from the manuscript. We made these calculations following Pruppacher and Klett (2010, doi: 10.1007/978-0-306-48100-0) and the results are displayed in the figure below. Two calculations are shown: The first is based on the simple assumption of the Stokes regime (blue in the figure) and the second is corrected for slip-flow effects, which become important for small particles (red in the figure). All particles are assumed to be spherical liquid droplets and the flow is assumed to be laminar. The two theoretical calculations are similar and agree with the velocities we derived from the probes. Specifically, the theoretical calculations indicate settling rates between 0 and 0.01 m/s during the first hour of the case when we estimated ~0.01 m/s using the probe data and 0.03-0.035 m/s during the mature phase of the fog when the probes suggested a settling rate of 0.03-0.04 m/s. We decided to show the new figure here to document our work to address this question, but have elected not to add it to the manuscript. Instead, we have added two sentences to the text explaining that the settling rates implied by the time series derived from the probes are consistent with theoretical calculations, referring readers to Pruppacher and Klett (2010).

- These sentences can be found on Lines 23-26 of Page 9.
- The Pruppacher and Klett reference now appears in the refrence list.

[Figure]

2. Although the Summit site is likely quite flat, can the authors rule out the possibility of any uplift effects when the wind direction and speed might favor adiabatic cooling based on the local but subtle topography.

This is an excellent question. Reviewer #1 asked a similar question, but referring to advection from more distant regions. The area around Summit Station is indeed quite flat. A number of detailed topographic surveys have been conducted in the area by the Cold Regions Research and Engineering Laboratory (CRREL) for logistical management of the station structures. For example, see https://www.geosummit.org/sites/default/files/docs/Summit-Station_ERDC-CRREL_TR-16-16.pdf. These surveys indicate that the maximum slope is approximately 1% at the kilometer scale. It is <1% towards the south, into the predominant wind direction from which our sampling was conducted. Locally around the station, variability in surface height is ~±2-3 m and is generally more complex than the surrounding environment because of localized drifting associated with station structures. Much of our analysis of the case studies assumes relative spatial homogeneity. Unfortunately, we cannot rule out the possibility that local topographic effects (or advection from more distant areas) might explain some of the observed time-evolution of the fogs at the location of the measurements. Therefore, we have included a statement acknowledging these limitations in the introductory remarks at the beginning of the case study section.

3. Is it possible that thermal tides, as measured by the surface barometer, might have a role in the observed diurnal signals when solar radiation is not the obvious forcing mechanism?

This is a very intriguing hypothesis! We have conducted some work linking buoyancy waves above the boundary layer to fluctuations in the thermal structure of the surface layer and associated modification of the fog microphysics. This was presented at the POLAR2018 conference in June 2018 in Davos, Switzerland, and is the subject of ongoing study (AC3-2010: http://www.professionalabstracts.com/POLAR2018/iPlanner/#/grid/1529539200). Only a small

portion of that work was incorporated into the current manuscript and can be found in the discussion of Figure 7. Because we have observed buoyancy waves to create areas of convergence and divergence at the surface (see Section 4.2) that also appear to modulate fog microphysics (discussed in the poster, but not in the manuscript), it is plausible that gravitational waves, such as thermal tides could be responsible and that such phenomena could trigger condensation. We revisited the POLAR2018 study and some additional cases and believe that while the occurrence of the buoyancy waves in the case study coincides with the diurnal cycle, this is incidental and not causal. We will certainly keep this idea in mind as a plausible candidate for the missing mechanism we highlight at the end of Section 4.2, but as yet we have not found supporting evidence.

[revised manuscript text omitted]

---

## Author Response (AR3)

Reviewer:

The article has been rewritten for clarity and is substantially more clear. Useful information is provided about the statistics of Greenland fogs, and important feature for understanding the radiation balance over the Greenland ice sheet. It is commendable the effort required to obtain these difficult measurements. I only have one minor suggestion which is that Figure 3 looks weird, and I think would look much more sensible if it were plotted on a log-linear scale. Negative exponentials drop of more quickly than power-laws, which is what seems to be the case in the data. Also, it looks like the binning is linear rather than logarithmic, so that would need to be corrected in any case.

From the author:

We have updated Figure 3 to use a log-linear scale. The binning was in fact logarithmic, although finely resolved. We have also adjusted the marker size to make the data more visually distinct for the new scale in both the plot and the legend. The revised figure is shown at the bottom of the page.

We have also made the following updates to the Acknowledgements and References:
- We have included an acknowledgement to the two anonymous referees who reviewed the manuscript.
- We have updated the DOIs associated with the data set.
- We have replaced "doi:" with "https://doi.org/" for in the References section.
- We have updated Edwards-Opperman et al. (2017) in review with the full citation information for the published article (Edwards-Opperman et al., 2018).
- We have added DOI information missing from the Reference list for Hoch et al. (2007) and Houghton (1985).